# In vitro reconstitution of chromatin domains shows a role for nucleosome positioning in 3D genome organization

Elisa Oberbeckmann [1,4] ✉, Kimberly Quililan[2,3,4], Patrick Cramer[1] & A. Marieke Oudelaar [2] ✉

Eukaryotic genomes are organized into chromatin domains. The molecular mechanisms driving the formation of these domains are difficult to dissect in vivo and remain poorly understood. Here we reconstitute *Saccharomyces cerevisiae* chromatin in vitro and determine its 3D organization at subnucleosome resolution by micrococcal nuclease-based chromosome conformation capture and molecular dynamics simulations. We show that regularly spaced and phased nucleosome arrays form chromatin domains in vitro that resemble domains in vivo. This demonstrates that neither loop extrusion nor transcription is required for basic domain formation in yeast. In addition, we find that the boundaries of reconstituted domains correspond to nucleosome-free regions and that insulation strength scales with their width. Finally, we show that domain compaction depends on nucleosome linker length, with longer linkers forming more compact structures. Together, our results demonstrate that regular nucleosome positioning is important for the formation of chromatin domains and provide a proof-of-principle for bottom-up 3D genome studies.

The spatial organization of the genome modulates nuclear processes, including transcription, replication and DNA repair. Eukaryotic genomes are organized into chromatin structures across different scales. The smallest unit of chromatin is the nucleosome core particle, which consists of 147 base pairs (bp) of DNA wrapped around a histone octamer[1–3]. Nucleosome core particles are connected by short DNA 'linkers' and form nucleosome arrays, which further organize into secondary structures that define the orientation of subsequent nucleosomes with respect to each other. At a larger scale, eukaryotic genomes organize into self-interacting domains. In mammals, these domains are formed by at least two distinct mechanisms[4]. First, active and inactive regions of chromatin form functionally distinct compartments that span a wide range of sizes[5]. Second, a process of loop extrusion, mediated by cohesin and CCCTC binding factor (CTCF), organizes the genome into local structures termed topologically

associating domains (TADs), which usually range from 100 kbp to 1 Mbp in size[6,7].

The higher-order organization of the genome into self-interacting domains is conserved in eukaryotes with smaller genomes, including *Drosophila melanogaster*[8] and *Saccharomyces cerevisiae*[9,10], in which domain sizes range from 10 to 500 kbp and 2 to 10 kbp, respectively. These domains are usually referred to with the general terms chromatin domain, chromosomal domain or chromosomal interaction domain[8,10]. This reflects that the nature of the domains in these species and the mechanisms by which they are formed are less well understood. Hereafter, we will therefore adopt the general term chromatin domain to refer to these domains. Because the boundaries of chromatin domains in fly[8,11,12] and yeast[10,13] frequently overlap with promoters of highly transcribed genes, it has been proposed that the process of transcription or the transcriptional state of chromatin

[1]Max Planck Institute for Multidisciplinary Sciences, Department of Molecular Biology, Göttingen, Germany. [2]Max Planck Institute for Multidisciplinary Sciences, Genome Organization and Regulation, Göttingen, Germany. [3]Present address: The Francis Crick Institute, London, UK. [4]These authors contributed equally: Elisa Oberbeckmann, Kimberly Quililan. ✉e-mail: elisa.oberbeckmann@mpinat.mpg.de; marieke.oudelaar@mpinat.mpg.de

are key determinants of chromatin organization. There is currently no conclusive evidence for cohesin-mediated loop extrusion during interphase in these species, but it is possible that this process also contributes to the basic organization of their genomes. Progress in our understanding of the conserved, core mechanisms that drive higher-order genome organization in eukaryotes is complicated by the close relationship and functional interplay between chromatin domains, chromatin state and transcription. The fundamental principles underlying the formation of three-dimensional (3D) chromatin structures are therefore difficult to disentangle in the complex in vivo nuclear milieu and remain poorly understood.

To address these limitations, we set out to reconstitute chromatin domains in vitro, in order to identify the molecular mechanisms that drive the formation of higher-order 3D chromatin structures in a controllable experimental setup. To this end, we used an in vitro system for yeast chromatin reconstitution and established a chromosome conformation capture (3C) approach to map folding patterns of in vitro chromatin at subnucleosome resolution. Using this unique approach, we show that reconstitution of regularly spaced and phased nucleosome arrays by the addition of purified transcription factors (TFs) and ATP-dependent chromatin remodelers drives higher-order nucleosome folding into chromatin domains with remarkable similarity to yeast genome organization in vivo. The boundaries of these domains correspond to the nucleosome-free regions (NFRs) at the TF binding sites. We find that the strength of these boundaries depends on a combination of nucleosome array regularity and NFR width. Our work therefore shows that the establishment of regularly spaced and phased nucleosome arrays surrounding NFRs is in principle sufficient to reconstitute in vivo-like chromatin domains in yeast. By extension, this suggests that neither loop extrusion nor transcription is a prerequisite for the formation of yeast chromatin domains. By comparing three different remodelers that set distinct nucleosome linker lengths, we also show that the compaction of chromatin domains depends on the linker length. 3D chromatin models generated with molecular dynamics (MD) simulations confirm these results and highlight the fundamental principles that drive chromatin domain formation. Together, our work identifies an underappreciated role for nucleosome positioning in the formation of chromatin domains in yeast, which may have important implications for our understanding of genome organization across eukaryotic species.

## Results

### An in vitro system to study chromatin domain formation

To study higher-order folding of in vitro-reconstituted chromatin, we adapted a previously established system to reconstitute *S. cerevisiae* chromatin in vitro[14,15] (Fig. 1a). As a DNA template, we used a genomic plasmid library covering *S. cerevisiae* chromosomes V–IX. Each of these plasmids contains an ~7 kbp backbone and an insert covering a fraction of the *S. cerevisiae* genome with an average length of ~10 kbp. Incubation of this plasmid library with purified recombinant *S. cerevisiae* histone octamers (Extended Data Fig. 1a) in high-salt conditions and overnight dialysis into a low-salt buffer leads to spontaneous assembly of nucleosomes. This reconstitution system has the important benefit that it allows for the generation of chromatin with high nucleosome density that resembles in vivo chromatin[16]. To further facilitate the formation of high nucleosome densities in vitro, we used negatively supercoiled plasmid DNA amplified in *Escherichia coli*, which is thought to propagate nucleosome assembly during salt gradient dialysis (SGD)[17,18].

The positioning of nucleosomes in SGD chromatin is solely directed by the DNA sequence and therefore irregular. To reconstitute regular nucleosome positioning, we incubated the SGD chromatin with purified, sequence-specific DNA-binding TFs and ATP-dependent chromatin remodelers (Extended Data Fig. 1a). As TFs, we used the general regulatory factors Abf1 and Reb1. These factors bind promoter regions and act as pioneer factors and transcriptional activators[19–21]

(Extended Data Fig. 1b). As chromatin remodelers, we used INO80, ISW2 and Chd1. These remodelers create regular nucleosome arrays at TF binding sites that resemble in vivo chromatin (Extended Data Fig. 1b). The nucleosomes in these arrays are evenly spaced and phased relative to reference sites. In our system these correspond to Abf1 and Reb1 binding sites (Extended Data Fig. 1b). INO80, ISW2 and Chd1 have an intrinsic 'ruler' function that determines the spacing between the nucleosomes in the reconstituted arrays. INO80 generates relatively large linkers, ISW2 forms medium-sized linkers and Chd1 forms relatively small linkers[14,22]. We also used the remodeling the structure of chromatin (RSC) complex, which does not have spacing activity but has an important role in maintaining NFRs (Extended Data Fig. 1b)[21,23]. For our experiments, we incubated the SGD chromatin with both TFs and one of the remodelers. As controls, we used chromatin incubated with the two TFs in the absence of the remodelers (TF only) and chromatin incubated with the remodelers in the absence of the TFs (no TF; Extended Data Fig. 1b,c). To confirm the binding of the TFs in all conditions in which they are present, we performed in vitro chromatin immunoprecipitation followed by sequencing (ChIP–seq) experiments for Abf1 and Reb1 (Extended Data Fig. 2).

To analyze the 3D folding patterns of reconstituted chromatin, we established a 3C approach that is compatible with our in vitro setup (Fig. 1a; Methods). 3C is based on digestion and subsequent proximity ligation of cross-linked chromatin and thereby allows for the detection of spatial proximity between DNA sequences using high-throughput sequencing[24–26]. An important aim of our study is to identify the folding pattern of individual nucleosomes, which requires 3C analysis at very high resolution. We therefore adapted the recently developed Micro-Capture-C (MCC) approaches, which are based on digestion with micrococcal nuclease (MNase) and support a resolution of 1–20 bp (refs. 27,28). To enable efficient analysis of the 3D conformation of SGD chromatin, we optimized the cross-linking, digestion and ligation conditions (Extended Data Fig. 3; Methods). After reverse cross-linking and purifying the ligated DNA, we sheared the DNA to ~200 bp and performed 150 bp paired-end sequencing to enable direct sequencing of the ligation junctions. Instead of using capture oligonucleotides to enrich a subset of the yeast genome, we used a plasmid library that covers only chromosomes V–IX, approximately a quarter of the yeast genome. This corresponds to ~3 Mb of DNA in total, which can be sequenced with high coverage and thus analyzed at very high resolution. We refer to the optimized procedure as in vitro Micro-C.

Because the in vitro Micro-C procedure allows for direct identification of ligation junctions, it is possible to define the orientation of the interacting nucleosomes based on the direction of the reads and to distinguish reads resulting from an inward interaction and reads resulting from regions that have never been digested (Fig. 1b; Methods). Comparison of in vitro Micro-C data of independent samples of remodeled chromatin shows a high degree of correlation between replicates, thus confirming the robustness of the in vitro Micro-C protocol (Fig. 1c). Notably, analysis of the interaction patterns of SGD chromatin shows strong enrichment of intraplasmid interactions compared to interplasmid interactions (Fig. 1d and Supplementary Table 2). This indicates that in vitro Micro-C predominantly captures meaningful higher-order interactions between nucleosomes contained within individual plasmids with minimal spurious ligation between different plasmids. To investigate whether these interaction patterns are influenced by topological constraints related to the circular nature of the plasmids, we also generated Micro-C data of linearized SGD chromatin. For these experiments, we added to the remodeling reaction the restriction enzyme BamHI, which has a 6 bp recognition motif and cuts on average on 1–2 sites per plasmid. As expected, we find that plasmid linearization creates additional interaction 'boundaries' at the BamHI restriction sites, because the regions upstream and downstream of digested restriction sites are no longer adjacent but on opposite ends of the linearized plasmids (Fig. 1d).

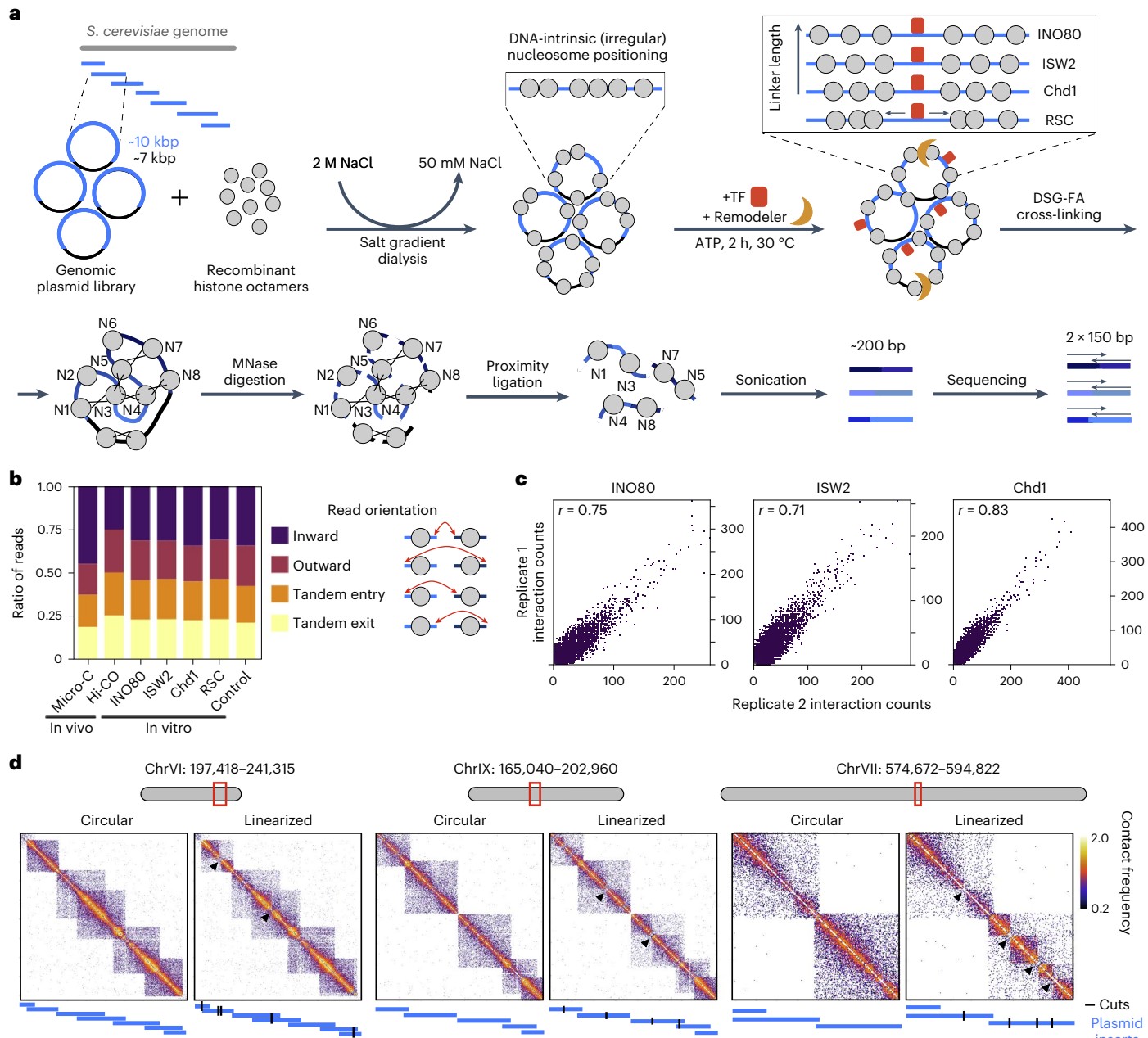

**Fig. 1 | An in vitro system to study higher-order chromatin structure.**
**a**, Schematic overview of the in vitro Micro-C procedure. **b**, Proportion of in vitro Micro-C read numbers clustered by the orientation of the interacting nucleosomes, as indicated in the scheme on the right. Chromatin used for in vitro Micro-C was incubated with the indicated remodeler and the TFs Abf1 and Reb1 or with the TFs only (control). For comparison, in vivo Micro-C[13] and Hi-CO[33] data are shown. **c**, Correlation plots of interaction frequencies of in vitro Micro-C replicates. Pearson's correlation coefficient *r* is indicated. **d**, Contact matrices displaying in vitro Micro-C data for three genomic regions. SGD chromatin was incubated with TFs and INO80 (circular) or with TFs, INO80 and the restriction enzyme BamHI (linearized). Plasmids are shown in blue; BamHI recognition sites are shown in black; arrowheads show sites that have been digested with BamHI, which leads to the separation of the domains. log₁₀ interaction counts are plotted at 80 bp resolution.

Note that there are still low-frequency interactions spanning these boundaries, as digestion is not complete (Extended Data Fig. 4a) and regions upstream and downstream of digested restriction sites are still connected on plasmids with one BamHI restriction site. However, notably, linearization of the plasmids does not affect nucleosome positioning (Extended Data Fig. 4b) or patterns of higher-order nucleosome folding (Fig. 1d and Extended Data Fig. 4c–e). For all remaining experiments, we therefore used circular plasmids. In addition, we used a small fraction of the digested sample for MNase-seq[29] to confirm efficient remodeling and regular in vivo-like nucleosome positioning for all experiments (Extended Data Fig. 1b,c).

## Regular nucleosome arrays form chromatin domains

To investigate the higher-order folding patterns of SGD chromatin in further detail, we plotted contact matrices of regions contained on individual plasmids at high resolution (40 bp). We excluded regions covered by multiple plasmids. The matrices therefore span 3 to 7 kbp. Because chromatin domains in yeast are on average 2–10 kbp in size, these matrices allow for a detailed comparison of domain organization in in vitro and in vivo chromatin[13] (Fig. 2).

As we only use 2 of 256 putative yeast TFs[30], we focused our analyses on regions with a high abundance of Abf1 and Reb1 binding sites. We find that the patterns of higher-order genome folding in such regions

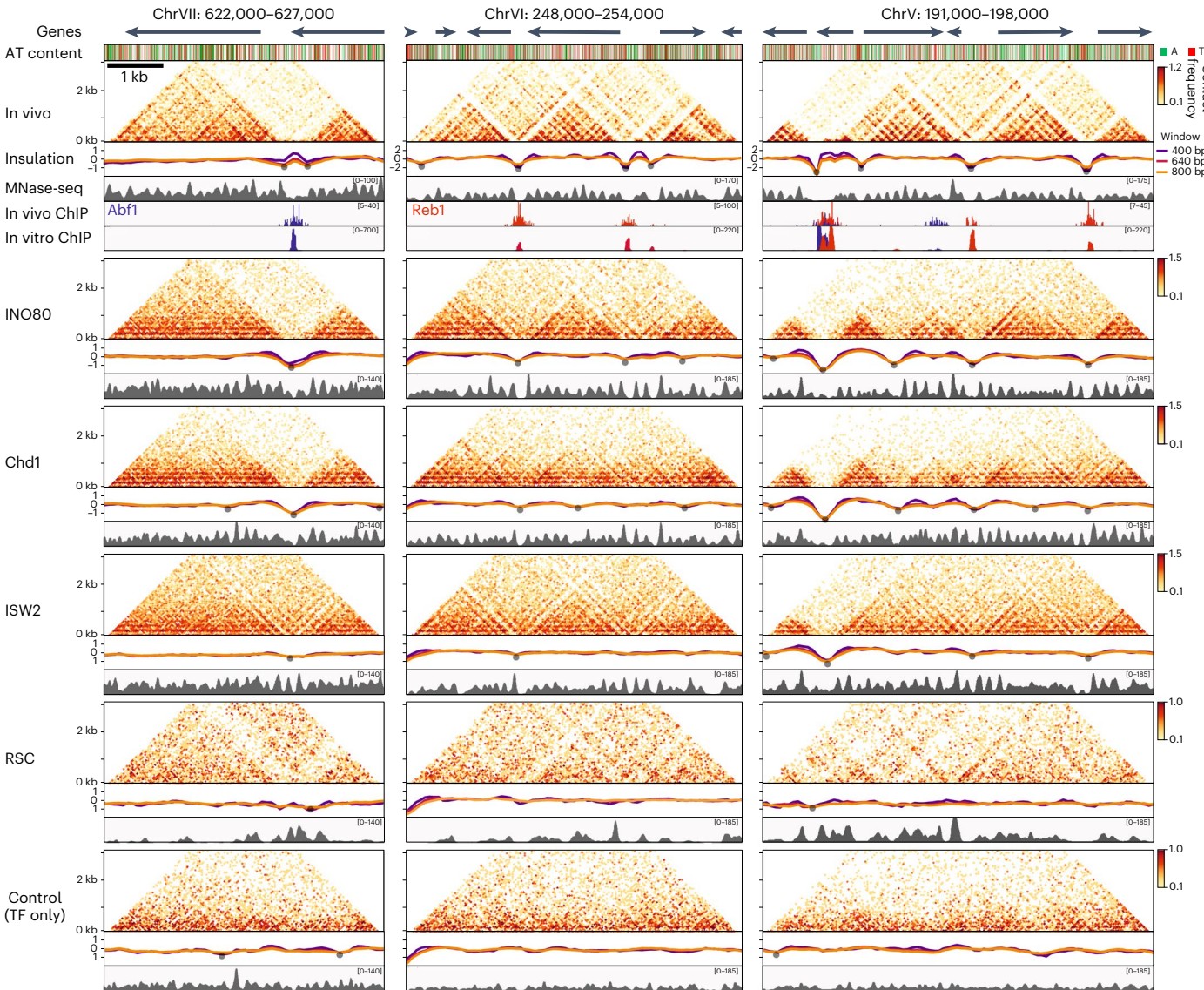

**Fig. 2 | Regularly spaced and phased nucleosome arrays surrounding NFRs are required and sufficient to reconstitute chromatin domains.** Contact matrices displaying in vivo[13] and in vitro Micro-C data with corresponding insulation scores and nucleosome occupancy profiles (MNase-seq) for three genomic regions. Gene (arrows) and sequence composition annotation are shown at the top (A, adenine; T, thymine). In vivo[19] and in vitro ChIP–seq data for Abf1 (blue) and Reb1 (red) are shown below the in vivo data. Chromatin used for in vitro Micro-C was incubated with the indicated remodeler and the TFs Abf1 and Reb1 or with the TFs only. Strongly insulating boundaries are labeled with gray dots. Micro-C data are plotted as $\log_{10}$ interaction counts at 40 bp resolution. Insulation scores are calculated at 80 bp resolution with three different sliding windows.

of in vitro chromatin are dependent on the chromatin remodeler used during reconstitution. Interestingly, the presence of chromatin remodelers that generate regular nucleosome arrays (INO80, Chd1 and ISW2) drives higher-order genome folding into chromatin domains with a striking similarity to domain organization in vivo. The boundaries of these domains correspond to the NFRs that are formed at Abf1 and Reb1 binding sites, as shown by the corresponding nucleosome occupancy patterns derived from the MNase-seq data. Although the occupancy and 3D organization of nucleosomes in in vitro chromatin in the presence of remodelers with spacing activity generally strongly resemble the in vivo chromatin, there are subtle differences. Most notably, we observe the formation of additional NFRs in in vitro chromatin, which correspond to additional domain boundaries (Fig. 2, right). These boundaries are associated with Abf1 and Reb1 motifs that are not bound in vivo but show clear enrichment in our in vitro ChIP–seq data. Conversely, we do not observe in vitro chromatin domains in regions that do not contain Abf1 and Reb1 binding motifs, even though such regions

may form domains in in vivo conditions, in which they are bound by other TFs (Extended Data Fig. 5).

To further investigate the requirements for chromatin domain formation, we performed several additional reconstitution experiments in which we omitted or exchanged the TFs and/or chromatin remodelers. We find that incubation of SGD chromatin with TFs only does not lead to specific 3D interaction patterns (Fig. 2), indicating that binding of TFs without remodeling of chromatin is not sufficient for the formation of domains. Incubation of SGD chromatin with chromatin remodelers in the absence of TFs does not result in domain formation either. This shows that the formation of NFRs at sites bound by TFs is important for the formation of chromatin domains (Extended Data Fig. 6). Finally, incubation with TFs and RSC, which leads to the formation of an NFR but not to regularly spaced nucleosomes surrounding the NFR, also does not lead to the formation of in vitro chromatin domains (Fig. 2), indicating that regular nucleosome spacing is also required. We, therefore, conclude that both regularly spaced and

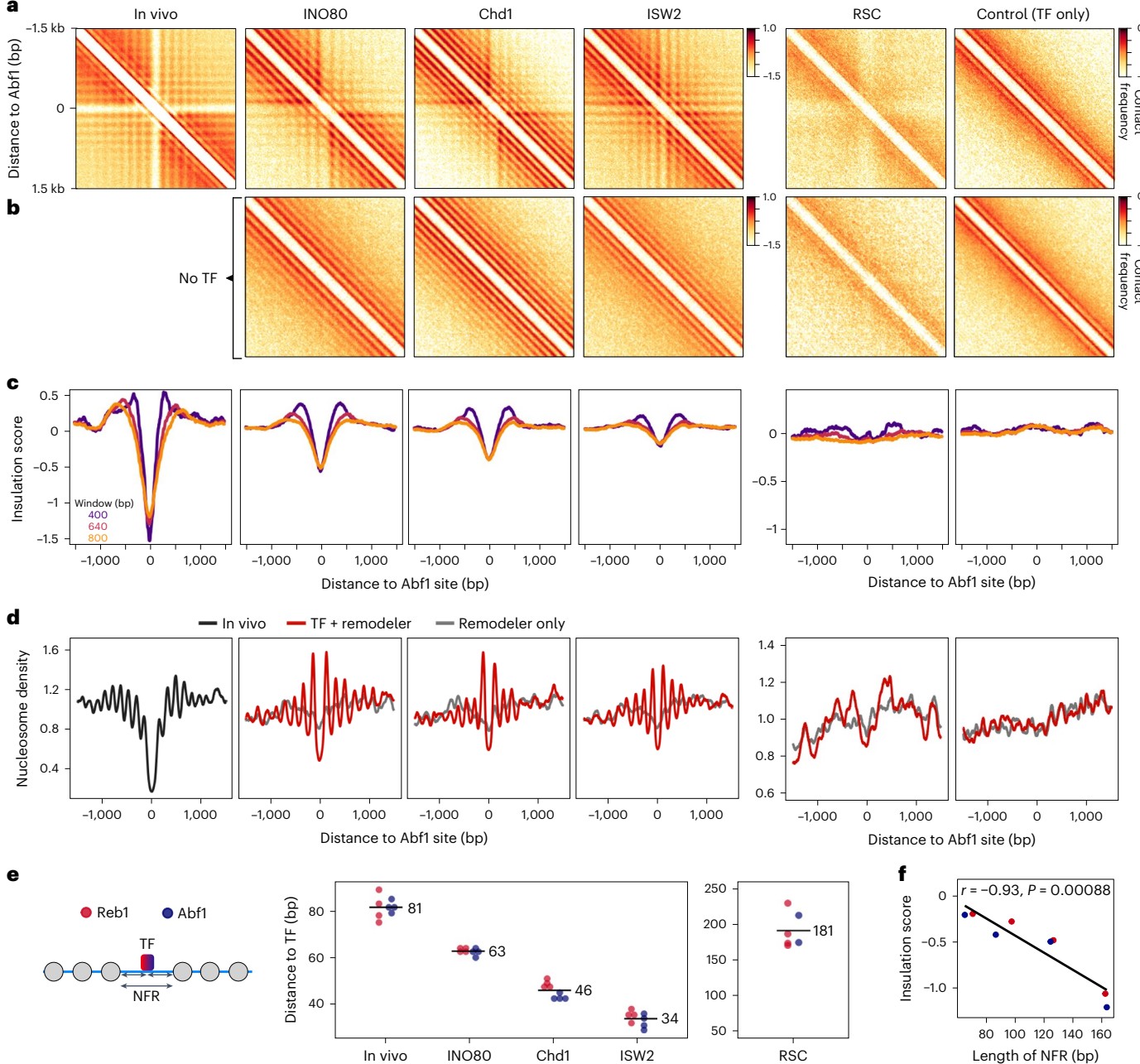

**Fig. 3 | NFR width correlates with the insulation strength of chromatin domains. a**, Pile-up analysis of contact matrices aligned at Abf1 binding sites. Chromatin was incubated with the indicated remodeler and the TFs Abf1 and Reb1 or with the TFs only. In vivo Micro-C data[13] are shown for comparison. $\log_{10}$ interaction counts are plotted at 20 bp resolution. **b**, Pile-up analysis of contact matrices as described in **a**. Chromatin was incubated with the indicated remodeler but without the TFs. **c**, Insulation scores derived from Micro-C data shown in **a** calculated at 80 bp resolution. Three different sliding windows are shown. **d**, Nucleosome occupancy profiles (MNase-seq) corresponding to **a** and **c**. In vivo chromatin is shown in black; in vitro chromatin incubated with a remodeler and TFs is shown in red; in vitro chromatin incubated with remodelers but without TFs is shown in gray. **e**, Distances from nucleosome border to TF calculated as indicated in the scheme on the left (small arrows). Values are derived from MNase-seq data as plotted in **d** and Extended Data Fig. 8d. Distances from two independent replicates ($n = 2$) and upstream or downstream direction of each replicate are shown. Numbers indicate mean distances. **f**, Correlation plot of insulation score minima at Abf1 (**c**) and Reb1 centers (Extended Data Fig. 8c) versus length of NFR. NFR lengths are calculated as shown in **e** (large arrow) and correspond from left to right to ISW2, Chd1, INO80 and in vivo data. Pearson's correlation $r$ and two-sided statistical significance $P$ are indicated. Minima are derived from insulation scores calculated with an 800 bp window.

phased nucleosome arrays surrounding NFRs are necessary for the formation of chromatin domains and that nucleosome positioning directly influences higher-order genome folding.

## NFR width determines the boundary strength of domains
To explore the relationship between TF enrichment and the formation of domain boundaries in a genome-wide manner, we correlated the

called boundaries with Abf1 and Reb1 ChIP–seq data (Extended Data Fig. 7). We find that regions with strong insulation are highly enriched for Abf1 and Reb1 binding sites, both in vivo and in vitro. To investigate the features of chromatin domain boundaries further, we performed pile-up analyses of the chromatin interactions in a 3 kbp region surrounding the TF binding sites in the in vivo and in vitro Micro-C data (Fig. 3a,b and Extended Data Fig. 8a,b). The resulting in vivo 'meta'

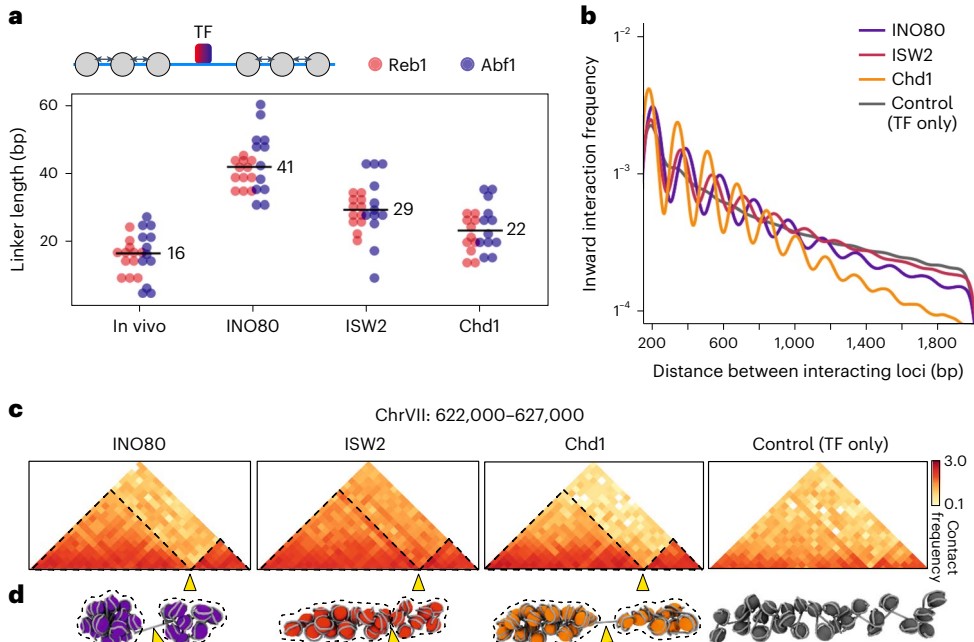

**Fig. 4 | Chromatin domain compaction depends on DNA linker length.**
**a**, Values for linker distances plotted as indicated in the scheme on the top (small arrows). Data are derived from MNase-seq as shown in Fig. 3d and Extended Data Fig. 8d. Three individual linker distances from upstream or downstream directions and from two independent replicates ($n = 2$) are plotted. Numbers indicate mean distances. **b**, Interaction frequency as a function of genomic distance plotted for inward-facing nucleosome interactions derived from in vitro Micro-C data from chromatin incubated with the indicated remodeler and the TFs Abf1 and Reb1 or with the TFs only. $\log_{10}$ interaction frequencies are plotted. **c**, Nucleosome-binned contact matrices displaying in vitro Micro-C data for the indicated region. **d**, MD simulations[33] of regions shown in **c**. Arrowheads point toward boundaries at NFRs corresponding to **c**. Stippled lines highlight chromatin domains corresponding to **c**.

contact matrices show three distinct interaction patterns: (1) a banding pattern parallel to the diagonal, which indicates regular spacing between nucleosomes; (2) horizontal and vertical bands that form a 'grid' of phased nucleosomes that are aligned to the NFR at the Abf1 and Reb1 binding sites and (3) insulation between the regions upstream and downstream of the TF binding sites.

The meta-contact matrices of in vitro chromatin reconstituted with remodelers with spacing activity (INO80, Chd1 and ISW2) closely resemble the in vivo matrices (Fig. 3a and Extended Data Fig. 8a). However, there are subtle differences in the interaction patterns that are established by these remodelers with respect to insulation and boundary strength (Fig. 3a,c and Extended Data Fig. 8a,c). INO80 generates the most distinct boundaries, characterized by strong insulation between the regions upstream and downstream of the Abf1 and Reb1 binding sites. Remodeling with Chd1 and ISW2 results in less insulation, with ISW2 creating the weakest domain boundaries. These differences are also apparent in the contact matrices from individual chromatin regions (Fig. 2). The meta-contact matrices also clearly show that reconstitution of chromatin in the presence of chromatin remodelers but in the absence of TFs results in regular nucleosome spacing without phasing, which is not associated with the formation of domain boundaries (Fig. 3b and Extended Data Fig. 8b).

To further investigate chromatin properties that regulate insulation strength, we compared the Micro-C data to MNase-seq meta profiles that show the distribution of nucleosomes in corresponding regions (Fig. 3d and Extended Data Fig. 8d). These data confirm that regular nucleosome positioning is required for the formation of chromatin domain boundaries, because chromatin incubated with remodelers that create regular nucleosome arrays (INO80, Chd1, and ISW2) forms strong domain boundaries, whereas incubation with RSC does not result in the formation of regular nucleosome arrays and clear domain boundaries. The MNase-seq data also show that of the three remodelers with spacing activity, INO80 forms the

widest NFR, followed by Chd1 and ISW2. This indicates that boundary strength is dependent on the size of the NFR established at TF binding sites. To confirm this, we calculated the distance between the nucleosome borders and TF binding sites to determine the width of the NFR (Fig. 3e,f). We find that INO80, Chd1 and ISW2 create NFRs of distinct sizes, which vary between 126, 92 and 68 bp in width, respectively. Plotting the average insulation score against these NFR sizes indicates that insulation strength scales proportionally with NFR width (Fig. 3f).

Consistent with the literature[15,19,21], we find that RSC does not drive the formation of regular nucleosome arrays, but does generate very wide NFRs (Fig. 3d,e). We find that this results in a chromatin interaction pattern characterized by a depletion of signal at the NFR itself and weak insulation at the NFR. We therefore conclude that insulation strength is dependent on both the width of the NFR at the chromatin domain boundary and the formation of a regularly spaced nucleosome array, phased to the boundary site. Strong insulation in vivo is likely facilitated by a complex interplay between several remodelers[31,32].

### Domain compaction depends on nucleosome linker length

The in vitro chromatin reconstituted with the different remodelers varies not only in NFR width but also with regard to nucleosome linker length. INO80, ISW2 and Chd1 create linker DNA of 41 bp, 29 bp and 22 bp, respectively (Fig. 4a). To explore how linker length relates to the general compaction of chromatin domains, we plotted the interaction frequencies as a function of genomic distance for chromatin reconstituted with these three remodelers (Fig. 4b and Extended Data Fig. 9a). These interaction decay curves show a similar pattern of nucleosome interactions as previously observed in in vivo Micro-C and Hi-CO data[10,13,33]. The interaction decay is steepest for Chd1, which indicates that interactions between nucleosomes that are separated by large distances (>1,000 bp) are relatively rare in Chd1-remodeled chromatin. Such long-range interactions are more frequent in chromatin

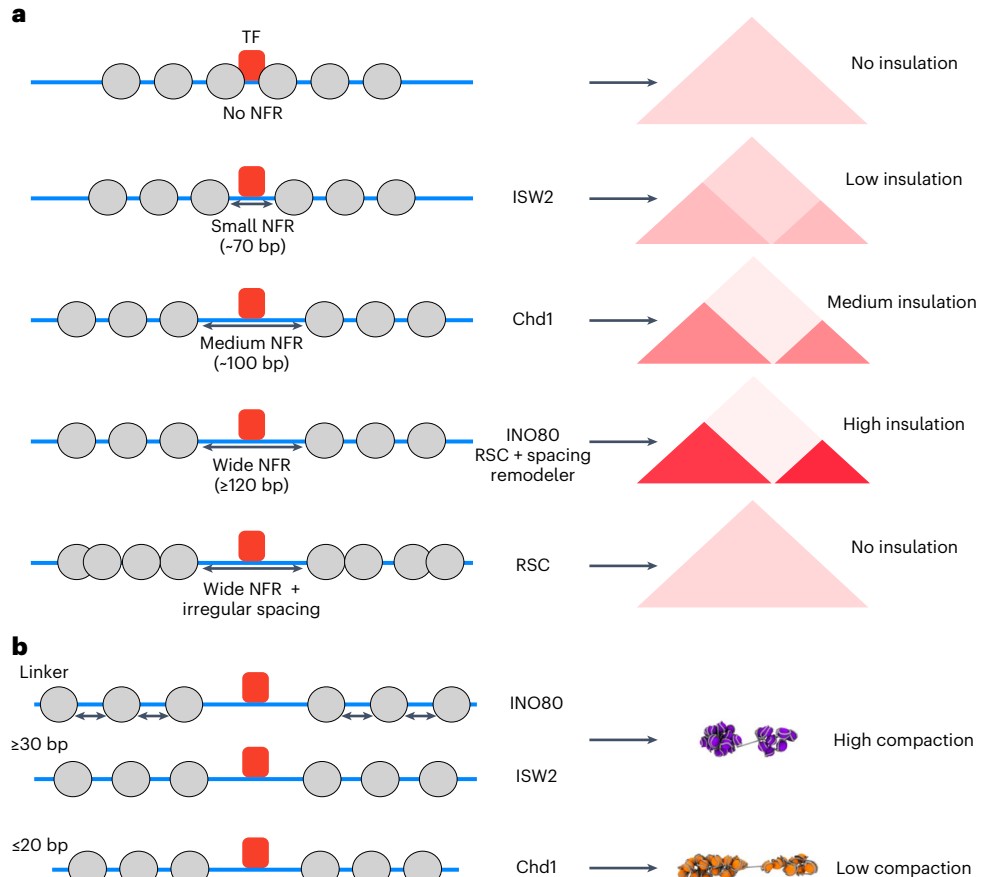

**Fig. 5 | Model for the mechanism of chromatin domain formation in *S. cerevisiae*. a**, Regular nucleosome positioning surrounding NFRs is required and sufficient for chromatin domain formation. The strength of the domain boundaries depends on the width of the NFR, which is generated in vivo by a combination of regulatory proteins, including sequence-specific TFs (for example, Abf1 and Reb1) and ATP-dependent chromatin remodelers (for example, INO80, ISW2, Chd1 and RSC). **b**, Compaction of chromatin domains depends on linker length, with longer linkers (≥30 bp) leading to more compaction compared to shorter linkers. Different linker lengths are generated in vivo by the action of various ATP-dependent chromatin remodelers.

remodeled with INO80 and ISW2. This suggests that nucleosomes with longer linker lengths form more compact chromatin. A possible explanation for these observations is that short linkers do not provide enough flexibility to support the folding of chromatin into very compact structures.

To further explore higher-order folding of in vitro chromatin, we performed MD simulations with a previously established pipeline[33]. These simulations are based on interaction data binned at nucleosome resolution and nucleosome orientation (Fig. 4c and Extended Data Fig. 9b). The resulting 3D models highlight the basic features of chromatin folding and their dependence on chromatin remodeling (Fig. 4d and Extended Data Fig. 9c). In the absence of remodelers, the nucleosomes do not form any specific, organized structures. However, in the presence of TFs and chromatin remodelers with spacing activity, the nucleosomes are organized in distinct domains that are separated by NFRs. Furthermore, comparison of the three remodelers highlights that INO80- and ISW2-remodeled chromatin with longer linkers is more compact compared to chromatin with shorter linkers that have been remodeled with Chd1. These data are consistent with previous studies that have reported a negative relationship between nucleosome linker length and chromatin compaction[34].

## Discussion

Our unique approach of combining reconstitution of native *S. cerevisiae* chromatin with high-resolution analysis of 3D genome structure at the level of chromatin domains has enabled us to gain important insights into the mechanisms underlying higher-order genome folding and the interplay between genome structure and function. We demonstrate that regular, in vivo-like nucleosome positioning is required and sufficient for the establishment of chromatin domains in yeast. By comparing the 3D organization of chromatin that has been reconstituted with different chromatin remodelers, we have also investigated how differences in basic chromatin features influence 3D genome folding. We find that the strength of domain boundaries is dependent on the width of NFRs (Fig. 5a) and that nucleosome linker length influences the compaction of chromatin domains (Fig. 5b).

The majority of in vitro domain boundaries are formed at Abf1 and Reb1 binding sites. However, we only observe the formation of chromatin domains when TF binding is accompanied by the formation of regularly spaced and phased nucleosome arrays by a chromatin remodeler with spacing activity (INO80, Chd1 or ISW2), which shows that TF binding alone is not sufficient for domain formation. This therefore indicates that the physical properties of histone–DNA interactions may drive the spontaneous formation of chromatin domains. We speculate that the separation of chromatin into two distinct domains is primarily driven by the DNA persistence length. To obtain interactions between two neighboring regions of chromatin, the DNA must bend by 180° or more. However, DNA is a very stiff polymer with a persistence length of 50 nm (~150 bp)[35]. Therefore, positively charged proteins, such as histone proteins, are required to neutralize the negative charge of DNA and enable DNA to bend[36]. This can explain why a stretch of naked DNA that is not bound by histone proteins can act as a stiff spacer and induce

strong insulation between two regions of chromatin. The formation of distinct domains on either side of this boundary is likely dependent on interactions between the regularly spaced nucleosomes[37].

The SGD reconstitution system that we have used in this study has the advantage that it allows for reconstitution of chromatin with in vivo-like features using a native, chromosome-wide DNA template[16]. However, this system also has limitations. Most notably, the reconstituted chromatin fibers are restricted in size, as they are generated from a genomic plasmid library with ~10 kbp inserts (Fig. 1a). This system is therefore limited to analyzing features of chromatin organization on a scale of ~10 kbp and cannot be used (in its current form) to analyze larger-scale features of genome folding. However, because the yeast genome folds into relatively small (2–10 kbp) domains in interphase[13], the SGD reconstitution system is suitable to study the formation of chromatin domains and the features of their boundaries in yeast. Furthermore, because Abf1 and Reb1 are the only TFs that are present in the reconstitution reactions, this system is limited to analyzing regions that are bound by these TFs and not informative for in vitro analysis of regions that are bound by other TFs in vivo.

The observation that regular nucleosome positioning drives the formation of in vivo-like chromatin domains in yeast has several implications that are relevant to the ongoing debate about the relationship between genome structure and function. For example, an important open question is whether transcription is responsible for the formation of chromatin domains or whether chromatin domains are formed first to enable transcription afterward. Because domain boundaries overlap with active gene promoters and boundary strength scales with increased RNA polymerase II binding in yeast[10,13], it has been suggested that transcription may have a driving role in the organization of the yeast genome. However, it is important to note that transcription and nucleosome positioning are closely connected because wide NFRs are a prerequisite for the formation of the pre-initiation complex (PIC) and PIC occupancy scales with increased NFR width[38–40]. Based on in vivo observations, it is therefore difficult to disentangle the role of transcription and nucleosome positioning in 3D genome organization. The strength of our in vitro approach is that we can uncouple these two processes. Although we have not directly investigated the function of transcription, the observation that we can reconstitute in vivo-like chromatin domains in the absence of the transcription machinery suggests that transcription is not required for domain formation in yeast per se. This is in agreement with computer simulations that have shown that nucleosome positioning alone can predict the domain organization of yeast interphase chromosomes[41]. Notably, transcription has also been implicated in the regulation of chromatin compaction, as previous studies in yeast have reported that active transcription is associated with higher compaction[10]. However, it is possible that this connection between transcription and chromatin compaction is mediated by associated changes in nucleosome spacing because highly transcribed genes generally have short linkers[42], which may be mediated by the recruitment of Chd1 to actively transcribed genes[22,43–45]. Transcription is also thought to have a role in the 3D organization of the genome of higher eukaryotes[46]. Although the relatively simple organization of the yeast genome is not directly representative of the more complex organization of mammalian genomes, it is interesting to point out that transcription and nucleosome positioning are similarly intertwined in mammals. We, therefore, speculate that it is possible that observations that have been interpreted as a driving role for transcription in genome folding in mammals could also reflect associated patterns of nucleosome positioning. Examples include the enrichment of active promoters at domain boundaries[47,48], which are also characterized by strong NFRs, as well as the interpretation of transcription perturbation experiments, which are generally also associated with changes in nucleosome positioning[38]. Although speculative, our findings therefore suggest that the role of transcription in genome organization in vivo may, to some extent, be mediated by the influence of transcription on nucleosome positioning.

Because we were able to reconstitute in vivo-like chromatin domains in the absence of cohesin, our experiments also indicate that loop extrusion is not per se required for basic domain organization of the yeast genome in interphase. It should be noted though that cohesin-dependent loop extrusion is thought to have an important role in the regulation of yeast chromatin architecture during S-phase and mitosis[49–52]. The notion that loop extrusion may not be required for the organization of the yeast genome during interphase is consistent with low levels of cohesin occupancy on chromatin during interphase[53] and with cohesin perturbation studies in yeast[49–52]. Together, this indicates that the chromatin domains that form in yeast interphase are not comparable to vertebrate TADs, which form via a loop extrusion process mediated by cohesin and CTCF[54,55] and may be more similar to compartments. However, it is interesting that the fine-scale interaction patterns at Abf1 and Reb1 binding sites in yeast are very similar to the patterns at CTCF binding sites in mammals[27,56]. We have previously reported that CTCF mediates local insulation (up to 10–20 kb) in mammalian genomes in a cohesin-independent manner[27]. In light of the findings in this study, it is likely that this is mediated by the strong influence of CTCF on nucleosome positioning[57–59]. In addition, it has been shown that perturbation of mammalian imitation switch remodeling complexes leads to changes in nucleosome positioning around CTCF binding sites that are accompanied by a decrease in TAD insulation and CTCF loops[60]. Together, these observations indicate that regular nucleosome positioning may have an important role in the formation of (local) chromatin boundaries across eukaryotic species. Although this process alone is sufficient to drive the basic organization of the small *S. cerevisiae* genome, higher eukaryotes have evolved additional mechanisms, including loop extrusion, to structure their larger genomes. The precise interplay between nucleosome positioning, loop extrusion, transcription and other mechanisms involved in higher-order genome folding is an exciting area to explore in more detail in future research.

## Online content

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

## Methods

### Ethical statement

This study is based on the use of insect, yeast and bacterial cells only and does not involve patients, tissue, multicellular organisms or mammalian cells.

### Genomic plasmid library

The genomic plasmid library used in this study covers ~3 Mbp of the *S. cerevisiae* genome, corresponding to chromosomes V–IX. It contains 384 clones and is part of a tiling plasmid library[61]. The library was expanded by transformation into chemically competent *dam−/dcm− E. coli* cells (Supplementary Table 1). For every 100 µl of competent *E. coli* cells, 2 µg of DNA was added for transformation. Transformed cells were grown on large lysogeny broth agar plates with kanamycin for 36 h at 37 °C; then colonies were combined into LB medium with 50 µg ml$^{-1}$ kanamycin and grown until $OD_{600}$ reached 2. Plasmids were extracted via a Plasmid Extraction Kit (Macherey–Nagel, PC2000).

### Expression and purification of yeast histone octamers

The strategy for purification of the yeast histone octamers[62] is described in detail in Supplementary Information.

### Expression and purification of the TFs Abf1 and Reb1

Abf1 or Reb1 containing expression plasmids[15] were expressed in *E. coli* BL21(DE3) RIL cells (Supplementary Table 1). In total, 2 l of terrific broth medium with 100 µg ml$^{-1}$ ampicillin and 20 µg ml$^{-1}$ chloramphenicol was inoculated with 100 ml of preculture and grown until OD 1. Cells were cooled down, protein expression was induced with 1 mM isopropyl-β-ᴅ-1-thiogalactopyranosid and continued at 16 °C for 16 h at 150 r.p.m. Cells were collected, resuspended in lysis buffer (20 mM Tris–HCl (pH 8 at 4 °C), 500 mM NaCl, 10 mM Imidazole, 1× Protease Inhibitor (0.284 µg ml$^{-1}$ leupeptin, 1.37 µg ml$^{-1}$ pepstatin A, 0.17 mg ml$^{-1}$ phenylmethylsulfonyl fluoride, 0.33 mg ml$^{-1}$ benzamidine)), flash frozen and stored at −80 °C.

For purification, cells were thawed at room temperature, incubated for 30 min on ice with 100 µg ml$^{-1}$ lysozyme and then sonicated for 1 min (10 s on and 10 s off), 50% peak power. Lysate was centrifuged for 1 h at 89,000*g*, 4 °C and filtered through a 0.45 µM filter. Cleared lysate was loaded onto a pre-equilibrated HisTrap HP 5 ml column (Cytiva), washed with 20 mM Tris–HCl (pH 8 at 4 °C), 200 mM NaCl, 20 mM Imidazole, 1× Protease Inhibitor and eluted with 20 mM Tris–HCl (pH 8 at 4 °C), 200 mM NaCl, 250 mM Imidazole, 1× Protease Inhibitor. Protein-containing fractions were pooled and loaded onto a HiTrap Q HP 5 ml column, washed and eluted with a gradient from 200 mM NaCl to 1 M NaCl. Protein-containing fractions were pooled, concentrated and subjected to gel filtration (Superdex 200 Increase 10/300 column (Cytiva) in 20 mM HEPES–NaOH (pH 7.5 at 4 °C), 300 mM NaCl, 10% glycerol, 1 mM dithiothreitol (DTT)), flash frozen and stored at −80 °C.

For in vitro ChIP–seq experiments, Abf1 and Reb1 proteins harboring a C-terminal Strep-tag II were purified. Reb1-Strep was purified as described previously[63]. Abf1-Strep-expressing cells were cultured and lysed as described above and then applied on a Strep-Tactin XT 4Flow 5 ml Column (IBA). After washing, protein was eluted with 50 mM biotin. Protein-containing fractions were pooled and loaded on a HiTrap Heparin 5 ml column (Cytiva). Protein was eluted with a gradient from 200 mM NaCl to 800 mM NaCl. Protein-containing fractions were pooled, concentrated, flash frozen and stored at −80 °C.

### Expression and purification of ATP-dependent chromatin remodeling enzymes

The strategies for purification of the chromatin remodelers Chd1, ISW2, INO80 (ref. 64) and RSC are described in detail in Supplementary Information.

### In vitro reconstitution of yeast chromatin

**SGD.** All steps were performed at room temperature. In total, 46 µg of plasmid library DNA were mixed with 40 µg of recombinant *S. cerevisiae* histone octamers in 400 µl assembly buffer (10 mM HEPES–NaOH (pH 7.6), 2 M NaCl, 0.5 mM EDTA (pH 8), 0.05% IGEPAL CA-630, 0.2 µg µl$^{-1}$ BSA). Samples were transferred to a Slide-A-Lyzer MINI dialysis cup (Thermo Fisher Scientific, 3.5 MWCO), which was placed in a 3 l beaker containing 300 ml high-salt buffer (10 mM HEPES–NaOH (pH 7.6), 2 M NaCl, 1 mM EDTA, 0.05% IGEPAL CA-630). Samples were gradually dialyzed against a total of 3 l low-salt buffer (10 mM HEPES–NaOH (pH 7.6), 50 mM NaCl, 0.5 mM EDTA, 0.05% IGEPAL CA-630) while stirring via a peristaltic pump over 12–14 h. After complete transfer of low-salt buffer, samples were dialyzed against 1 l low-salt buffer for 1 h. The chromatin was centrifuged for 1 min at 9,930*g* and the supernatant was transferred to a fresh protein low-binding tube. We refer to the in vitro-reconstituted chromatin as 'SGD chromatin.' SGD chromatin was stored for a maximum of 4 weeks at 4 °C.

**Remodeling of SGD chromatin.** Remodeling reactions contained the following: (1) 30 µl of SGD chromatin (2) TFs Abf1 and Reb1 and (3) one of the chromatin remodeling enzymes, in remodeling buffer (30 mM HEPES–KOH (pH 7.6), 3 mM MgCl$_2$, 2.5 mM ATP, 1.25 mM TCEP, 0.4 mM DTT, 10 mM creatine phosphate, 0.25 mM EGTA, 0.15 mM EDTA, 80 mM KOAc, 15 mM NaCl, 0.015% IGEPAL CA-630, 16.5% glycerol, 60 µg ml$^{-1}$ BSA, 10 µg ml$^{-1}$ creatine kinase) in a total reaction volume of 100 µl. To generate regularly spaced and phased nucleosome arrays, the SGD chromatin was incubated with the TFs, Abf1 and Reb1 (final concentration of 50 nM each), and one of the following remodeling enzymes: INO80, Chd1, ISW2 or RSC at a final concentration of 24 nM or 12 nM for RSC. Chd1-remodeling was performed in a remodeling buffer containing only 50 mM KOAc. For controls, the SGD chromatin was incubated with the TFs Abf1 and Reb1 alone (TF only) or with remodeling enzymes without the TFs (no TF). For the linearization experiments, 750 units of BamHI were added to a 100 µl remodeling reaction.

### In vitro Micro-C

In vitro Micro-C experiments were performed for two independent replicates per experimental condition, with exception of the cross-linking optimization experiments, for which single replicates were used.

To optimize cross-linking conditions, several conditions using formaldehyde (FA) only or FA in combination with disuccinimidyl glutarate (DSG) were tested (Supplementary Table 2). Cross-linking with FA only did not allow for efficient detection of long-range interactions between nucleosomes, irrespective of the FA concentration used. Addition of DSG drastically improved the detection of long-range interaction patterns, without affecting nucleosome positioning (Extended Data Fig. 3a–d).

In the optimized procedure, remodeled SGD chromatin was first incubated with DSG at a final concentration of 0.75 mM for 20 min. Then, FA was added at a final concentration of 0.05% for another 10 min. All cross-linking steps were performed at 30 °C. The cross-linking reactions were quenched with 10× Quenching Buffer (100 mM Tris (pH 7.5), 80 mM aspartate, 20 mM lysine) for 15 min. Cross-linked chromatin was diluted in digestion buffer (10 mM Tris–Cl (pH 7.6) and 1.7 mM CaCl$_2$) and supplemented with 0.4 Ku µl$^{-1}$ (final concentration) of MNase (NEB). The reaction was terminated with 2× Stop Buffer (10 mM Tris–HCl (pH 7.5), 20 mM EDTA, 400 mM NaCl, 2 mM EGTA) for 5 min.

The cross-linked and MNase-digested chromatin was purified using an Amicon Ultra 50 kDa centrifugal filter to remove <100 bp DNA fragments. The chromatin was diluted with low-salt buffer (20 mM HEPES–NaOH, 50 mM NaCl, 0.5 mM EDTA, 0.05% IGEPAL CA-630), loaded into the filter and the reaction was centrifuged at 2,000*g* for 2 min at room temperature. The reaction was washed two more times and concentrated to ~100 µl.

The proximity ligation reaction was adapted from the previously described MCC protocol[28], in which DNA end repair, phosphorylation and ligation are performed in a single tube. Nucleosomes were resuspended in T4 Ligation buffer supplemented with 0.4 mM dNTP, 2.5 mM EGTA, 20 U ml$^{-1}$ T4 polynucleotide kinase, 10 U ml$^{-1}$ DNA polymerase I large (Klenow) fragment and 30 U ml$^{-1}$ T4 DNA ligase in a total reaction volume of 400 µl. The reaction was incubated at 37 °C for 2 h followed by 22 °C for 8 h at 300 r.p.m. on an Eppendorf Thermomixer. The chromatin was reverse cross-linked with the addition of 1 mg ml$^{-1}$ Proteinase K (Life Technologies) and 0.5% SDS at 65 °C for >16 h. For DNA extraction, the reaction was supplemented with glycogen and subjected to ethanol precipitation. Digestion and ligation efficiencies were assessed using Fragment Analyzer. A successful ligation is indicated by the increase in fragment sizes of >320 bp (Extended Data Fig. 3e). Unligated mono-nucleosome fragments were further removed by purification with AMPure XP beads in a 0.9:1 ratio.

Approximately 150–200 ng of each library was sonicated to a mean fragment size of 200 bp using a Covaris S220 Ultrasonicator (peak incident power 175; duty factor 10%; cycles per burst 200; treatment time 250 s) followed by purification with AMPure XP beads in a 1.8:1 ratio. A total of 170–200 ng of the library was indexed using the NEBNext Ultra II DNA Library Prep Kit for Illumina. The manufacturer's protocol was followed with the following deviations: to maximize library complexity and yield, the PCR was performed in duplicate per ligation reaction using Herculase II reagents (Agilent Technologies). The parallel library preparations and PCR reactions were subsequently pooled for each reaction. The quality of the library and the molar concentration were measured using Fragment Analyzer and Qubit. The material was sequenced on Illumina NextSeq 550 with 150 bp paired-end reads, with each replicate having ~30 million reads.

### In vitro MNase-seq

During the in vitro Micro-C procedure, 10 µl cross-linked, MNase-digested and purified chromatin was used to assess the quality of MNase digestion. After digestion, DNA was extracted by ethanol precipitation. Fragment sizes were evaluated on a Fragment Analyzer. Approximately 15–20 ng of the digestion control was indexed using the NEBNext Ultra II DNA Library Prep Kit following the manufacturer's protocol and sequenced on Illumina NextSeq 550 with 40 bp or 150 bp paired-end reads.

### In vitro ChIP–seq

In vitro ChIP–seq experiments were performed for two independent replicates per experimental condition. Remodeling reactions with Strep-tagged Abf1 and Reb1 were set up as described above. After 2 h of incubation, chromatin was cross-linked with 0.05% FA, quenched and digested with MNase to obtain mainly mono-nucleosomes. Digested chromatin was incubated for 1–3 h with 10 µl prewashed MagStrep 'type3' Strep-Tactin beads (IBA) rotating at 4 °C. Afterward, beads were washed 4× with 500 µl Tris-buffered saline, 0.1% Tween rotating for 5 min at 4 °C. Then, DNA was eluted with 1× Stop Buffer (20 mM Tris–HCL (pH 7.5), 0.3% SDS, 10 mM EDTA, 150 mM NaCl, 125 ng µl$^{-1}$ glycogen, 5% Proteinase K (NEB)) and reverse cross-linked for 2 h at 65 °C. DNA was precipitated, and libraries were prepared with the NEBNext Ultra II DNA Library Prep Kit following the manufacturer's protocol and sequenced on Illumina NextSeq 550 with 40 bp paired-end reads.

### Reference datasets

In vivo Micro-C data[13] were downloaded from Gene Expression Omnibus (GEO; GSM2262329, GSM2262330 and GSM2262331) and re-analyzed using HiC-Pro[65]. Briefly, reads were mapped to the SacCer3 reference genome using Bowtie2 (ref. [66]), and valid pairs were extracted. The list of valid pairs for replicates was merged.

ChIP-exo data[19] (Abf1: GSM4449154; Reb1: GSM4449823) were mapped against the SacCer3 genome using Bowtie[67]. The 5′ ends of

reads were used and extended to 3 bp. Coverage files were calculated in R using GenomicAlignments[68] and visualized in the integrated genome viewer[69] (igv 2.8.6) for single loci analysis. For alignments, the corresponding ChIP-exo peaks from http://www.yeastepigenome.org/ were intersected with corresponding position weight matrix motifs that were previously generated[14]. This resulted in 119 sites for Abf1 and 128 sites for Reb1 on chromosomes V–IX.

### Analysis of in vitro Micro-C data

**Mapping.** Analysis was performed using the MCC pipeline as previously described[28]. Briefly, adapter sequences were removed using Trim Galore (Babraham Institute) and paired-end reads were reconstructed using FLASH[70]. Reconstructed reads were mapped to the ~3 Mbp plasmid library with the nonstringent aligner BLAT[71]. Uninformative reads (for example, plasmid backbone) were discarded, while the mapped reads in the FASTQ files were further mapped to the SacCer3 reference genome using Bowtie2 (ref. [66]). The aligned reads were then processed to identify the ligation junction and remove PCR duplicates using the MCC pipeline available from https://process.innovation.ox.ac.uk/software/p/16529a/micro-capture-c-academic/1. The MCC pipeline generates a list of the chimeric pairs, the base pair coordinates of the ligation junction and the direction of the read (upstream or downstream of the ligation junction). The numbers of reads for the subsequent analysis stages are summarized in Supplementary Tables 2 and 3. Interactions that are interchromosomal and less than 147 bp apart were removed from the pairwise interaction list. The read-pair orientation was classified into four groups based on the direction of the read relative to the junction (inward, outward, tandem entry and tandem exit) as described previously[33]. The ligation junction was further shifted by 80 bp to the nucleosome dyad.

**Contact matrices.** To generate contact matrices, chimeric pairs were aggregated in the cooler format using the cooler package[72] at 20, 40 and 80 bp resolution. Pearson correlation of 80 bp resolution contact matrices of replicate 1 and 2 was calculated using the hicCorrelate function of hiCExplorer package[73]. The chimeric pairs list for the two replicates was combined, and these data are represented throughout the manuscript. To create the nucleosome-binned contact matrix, we assigned each of the interacting pairs to one of the corresponding 66,360 nucleosome loci previously determined through MNase-seq[74].

**Pile-up analysis of contact matrices.** Contact frequencies with 20 bp resolution were extracted within a 3,000 bp window around Abf1 and Reb1 binding sites[19] and then aggregated and visualized using the cooltools package[75].

**Insulation scores and boundary calling.** Diamond insulation scores for 80 bp resolution contact matrices were calculated with different window sizes (400, 640 and 800 bp) using cooltools[75]. Insulating loci were called using a local minima detection procedure based on peak prominence. Highly insulating regions that correspond to strong boundaries were called according to the thresholding method 'Li' from the image analysis field[76]. Insulation scores were averaged in a 3,000-bp window around TF binding sites derived from either in vivo[19] or in vitro ChIP–seq data.

**Interaction decay curves.** Interaction counts were log$_{10}$ transformed and plotted as a function of genomic distance for each read-pair orientation.

### MD simulation

3D chromatin models derived from the in vitro Micro-C data were generated from simulated annealing–MD simulation as previously described[33,77]. In brief, the simulation represents nucleosomes as composed of histone and DNA beads, equivalent in number to the naturally

occurring nucleosome loci. A histone octamer corresponds to four histone beads having a 3 nm radius[78]. One bead of DNA corresponds to 5.88 bp, and a total of 23 sequential DNA beads is wrapped around four histone particles in a left-handed super-helical geometry for 1.65 turns with a radius of 4.18 nm and a pitch of 2.39 nm (ref. [78]).

## Analysis of MNase-seq data

For samples sequenced with 150-bp paired ends, sequencing reads were trimmed to 75 bp. For 40-bp paired-end reads, trimming was omitted. Reads were mapped with Bowtie[67] to the *S. cerevisiae* genome SacCer3 omitting multiple matches. Mapped data were imported into R Studio using GenomicAlignments[68] and only fragments with 125–205 bp length were kept. Nucleosome dyad length was reduced to 50 bp and smoothed by a 20 bp rolling window. Genome coverage was calculated. For single-gene visualization, data were converted to a GenomicRanges format, exported as bigwig file and loaded into the integrated genome viewer[69] (igv 2.8.6).

Views of coverage files were generated with a 2,001 bp or 3,001 bp window around in vivo + 1 nucleosome positions[14] or Abf1 and Reb1 binding sites[19], respectively. Only sites within chromosomes V–XI were used (1,184 sites for in vivo + 1 nucleosome, 119 sites for Abf1 and 128 sites for Reb1). Nucleosome signal was normalized per window.

Windows around in vivo + 1 nucleosome sites were sorted by Abf1 and Reb1 signal in the NFR. To this end, Abf1 (GSM2916412) and Reb1 (GSM2916410) ChIP–seq data[79] were merged, aligned as described above to in vivo + 1 nucleosomes and sorted by decreasing signal strength in a 180 bp window 160 bp upstream of the in vivo + 1 nucleosome site.

For composite plots, mean of normalized nucleosome signal was calculated and plotted for TF-bound genes (top 20% of genes), TF-unbound genes (bottom 80% of genes) or Abf1 and Reb1 binding sites.

Linker and NFR values were calculated as described previously[14]. In brief, peaks of nucleosome occupancy profiles around Abf1 and Reb1 binding sites were called. Then, the nucleosome repeat length was calculated (distance from peak to next neighboring peak) and 147 bp was subtracted to obtain the linker length of the first three nucleosomes upstream and downstream of the TF binding site. For the calculation of the distance to the TF binding site, the peak of the first nucleosome was called, the distance to the alignment point was calculated and 73 bp were subtracted.

## Analysis of in vitro ChIP–seq data

Reads were mapped with Bowtie[67] to the *S. cerevisiae* genome SacCer3 omitting multiple matches. Mapped data were imported into R Studio using GenomicAlignments[68] and only fragments smaller than 91 bp were kept. Genome coverage was calculated. For single-gene visualization, data were converted to a GenomicRanges format, exported as bigwig file and loaded into the integrated genome viewer[69] (igv 2.8.6).

Views of coverage files were generated with a 2,001 bp or 3,001 bp window around in vivo + 1 nucleosome positions[14] or Abf1 and Reb1 binding sites[19], respectively. The ChIP–seq signal was normalized for sequencing depth.

Windows around in vivo + 1 nucleosome sites were sorted by in vivo Abf1 and Reb1 ChIP–seq signal in the NFR. To this end, Abf1 (GSM2916407) and Reb1 (GSM2916412) ChIP–seq data[79] were aligned as described above to in vivo + 1 nucleosomes and sorted by decreasing signal strength in a 180 bp window 160 bp upstream of the in vivo + 1 nucleosome site.

For composite plots, the mean of the normalized signal was calculated and plotted for TF-bound genes and TF-unbound genes.

ChIP–seq peaks were called in each remodeler condition separately using MACS2 (ref. [80]) (2.2.8) with default settings. Then, all called peaks were merged and motif discovery was performed using MEME[81] (5.5.2). Only peaks with motif sites were used for the correlation with boundary sites.

## Statistics and reproducibility

We used two independent sets of reconstituted SGD chromatin samples to perform MNase-seq, in vitro Micro-C and in vitro ChIP–seq experiments. To correlate the two datasets, Pearson correlation coefficients were calculated.

No statistical methods were used to predetermine sample sizes, but our sample sizes are similar to those reported in previous publications[14,15]. Data collection was not randomized, as the researchers needed to know the experimental conditions to perform the experiments successfully. Data collection and analysis were not performed blind to the conditions of the experiments. No data were excluded from the analysis.

## Reporting summary

Further information on research design is available in the Nature Portfolio Reporting Summary linked to this article.

## Data availability

All raw sequencing data and processed data generated in this study are available for download at http://www.ncbi.nlm.nih.gov/geo/ via GEO accession GSE220647. In vivo Micro-C data are available via GEO accession GSM2262329, GSM2262330 and GSM2262331, and in vivo ChIP–seq data for Abf1 and Reb1 via GEO accession GSM4449154, GSM4449823, GSM2916412 and GSM2916410. Source data are provided with this paper.

## Code availability

The code used for MNase-seq and ChIP–seq data analysis is available on Zenodo via https://zenodo.org/records/10361971 (ref. [82]). The code used for Micro-C data analysis and molecular dynamics simulations is available on Zenodo via https://zenodo.org/records/10373300 (ref. [83]). The Micro-C code has been adapted from code available on https://github.com/jojdavies/Micro-Capture-C and can be used with scripts that are freely available for academic use on the Oxford University Innovation software store via https://process.innovation.ox.ac.uk/software/p/16529a/micro-capture-c-academic/1 (ref. [28]).

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

## Acknowledgements

We thank L. Farnung (former member of the Cramer laboratory) for sharing purified Chd1, plasmids and advice regarding protein purification; P. Korber (LMU Munich) for sharing the yeast plasmid library and Abf1/Reb1 expression plasmids; C. Kurat (LMU Munich) and J. Diffley (Francis Crick Institute) for sharing the INO80-FLAG strain; M. Singleton (Francis Crick Institute) for sharing *S. cerevisiae* histone co-expression plasmids; A. Galitsyna for support with analysis of Hi-CO data; and M. Ohno and Y. Taniguchi for support with molecular dynamics simulations. We are grateful to C. Dienemann, M. Lidschreiber, J. Söding and J. Walshe for providing feedback on the manuscript. We also thank all members of the Oudelaar and Cramer laboratories for helpful discussions. This work was supported by the MSc/PhD program 'Molecular Biology'—International Max Planck Research School at the Georg August University Göttingen (to K.Q.); the Max Planck Society (to P.C. and A.M.O.); the European Research Council (advanced grant CHROMATRANS, 882357 to P.C.) and the Deutsche Forschungsgemeinschaft (DFG) via SFB 1565 (project 469281184, project P02 to A.M.O.).

## Author contributions

E.O. and A.M.O. conceived, planned and supervised the project. E.O. and K.Q. performed experiments and analyzed data. P.C. provided conceptual advice. E.O. and A.M.O. wrote the manuscript, with input from K.Q. and P.C. P.C. and A.M.O. acquired funding.

## Funding

## Competing interests

The authors declare no competing interests.

## Additional information

**Extended data** is available for this paper at https://doi.org/10.1038/s41588-023-01649-8.

**Correspondence and requests for materials** should be addressed to Elisa Oberbeckmann or A. Marieke Oudelaar.

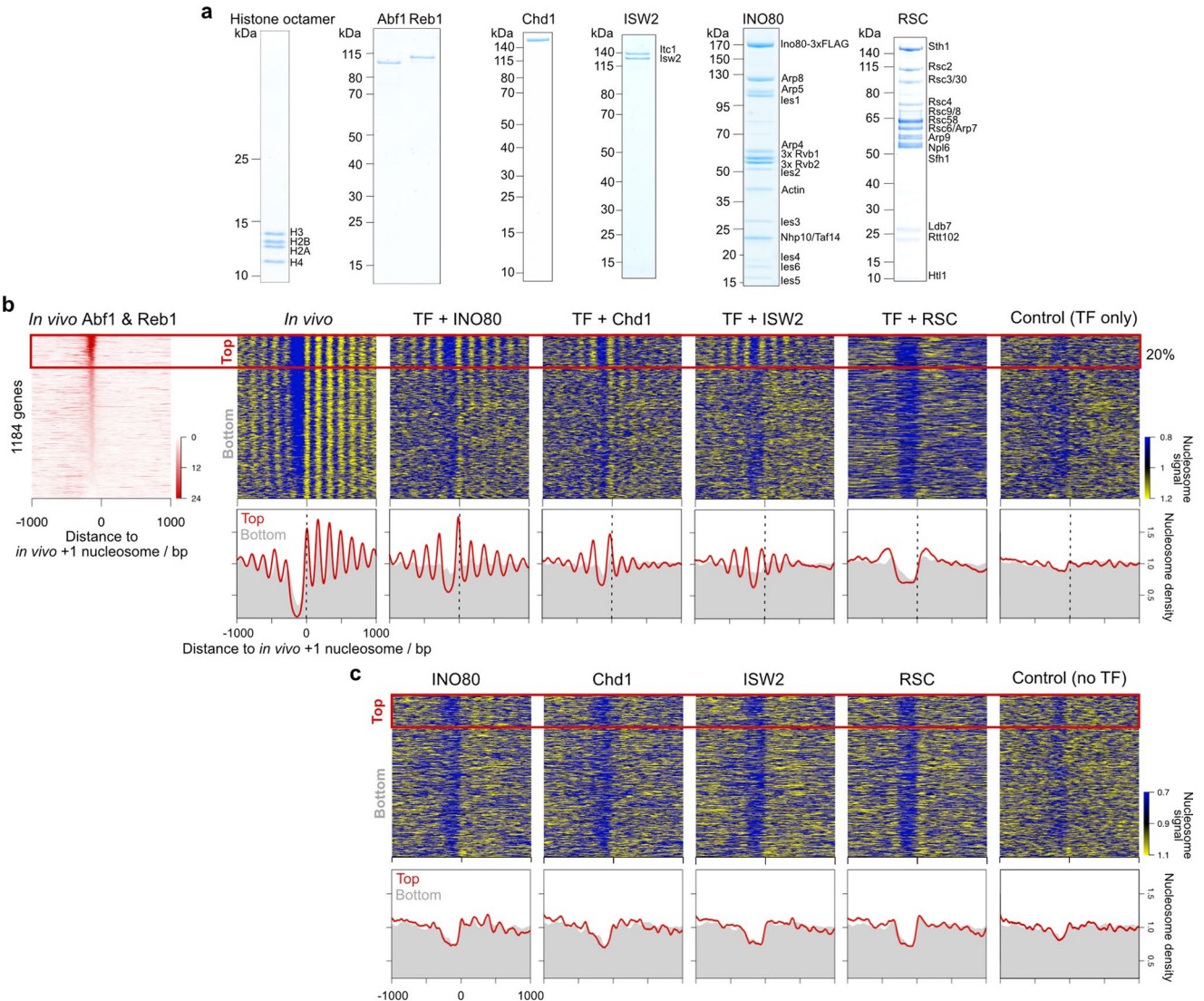

**Extended Data Fig. 1 | Reconstitution of native nucleosome positioning with purified proteins. a**, Representative example of SDS-PAGE analysis of purified proteins used for *in vitro* chromatin reconstitution. **b**, The left heatmap shows merged Abf1 and Reb1 SLIM-ChIP data[79] aligned at *in vivo* + 1 nucleosome positions and sorted by TF binding signal in NFRs (left). The right heatmaps (top) and composite plots (bottom) show nucleosome occupancy of reconstituted chromatin incubated with the indicated remodeler and the TFs Abf1 and Reb1 or with the TFs only. The genes in the heatmap are sorted according to TF binding as shown on the left. *In vivo* nucleosome positioning is derived from Micro-C data[13]. Composite plots show averaged nucleosome occupancy of TF bound genes (red, corresponding to top 20% of genes) or TF unbound genes (gray, corresponding to bottom of heatmap). **c**, Heatmaps as shown in panel **b** for chromatin incubated with the indicated remodeler in absence of TFs.

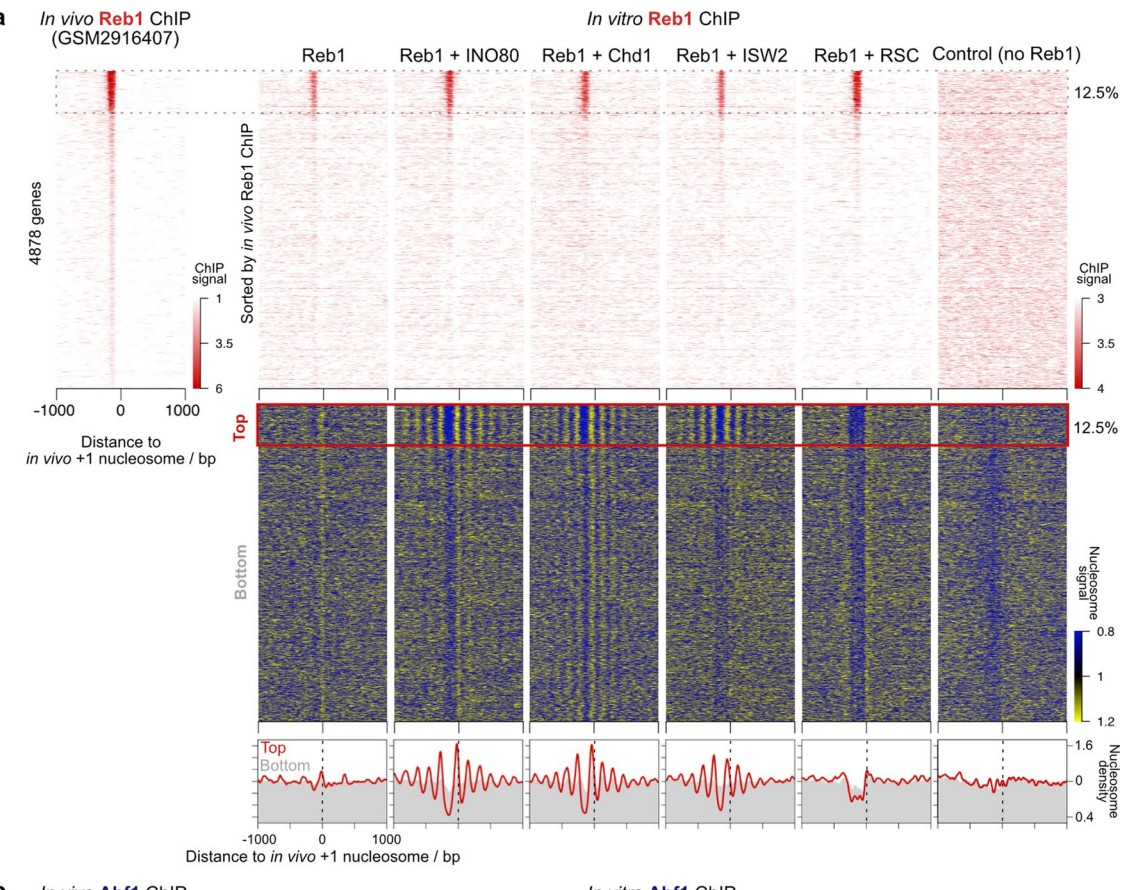

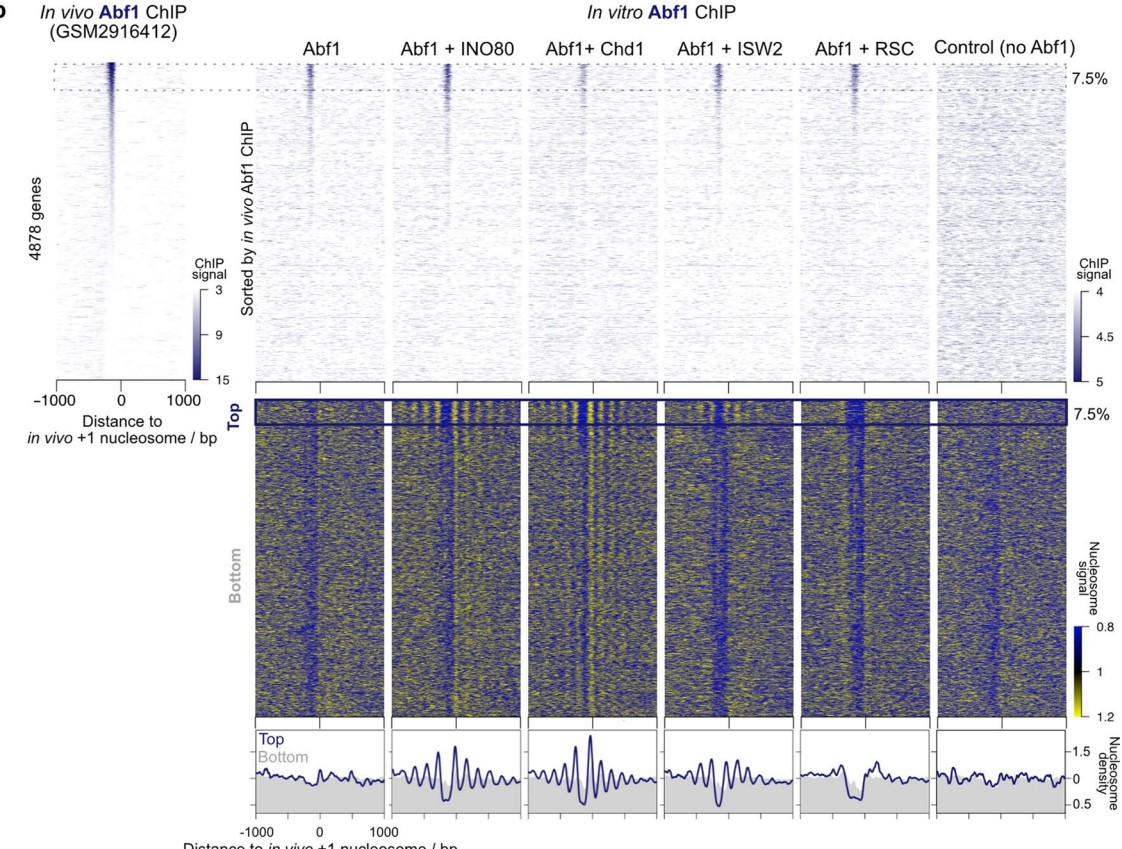

**Extended Data Fig. 2 | See next page for caption.**

**Extended Data Fig. 2 | ChIP-seq analysis of transcription factor binding to** *in vitro* **chromatin. a**, The left heatmap shows *in vivo* Reb1 SLIM-ChIP data[79] aligned at *in vivo* +1 nucleosome positions and sorted by TF binding signal in NFRs. The top-right heatmaps show *in vitro* Reb1 ChIP-seq data of reconstituted chromatin incubated with the indicated remodeler and the TF Reb1 or without remodeler and Reb1. The bottom-right heatmaps and composite plots show the corresponding MNase-seq data (input). The genes in all heatmaps are sorted according to TF binding as shown on the left. Composite plots show the averaged nucleosome occupancy of Reb1-bound genes (red, corresponding to top 12.5% of genes) or TF unbound genes (gray, corresponding to bottom of heatmap). **b**, Analysis as described in panel **a** for *in vivo* and *in vitro* Abf1 ChIP-seq data.

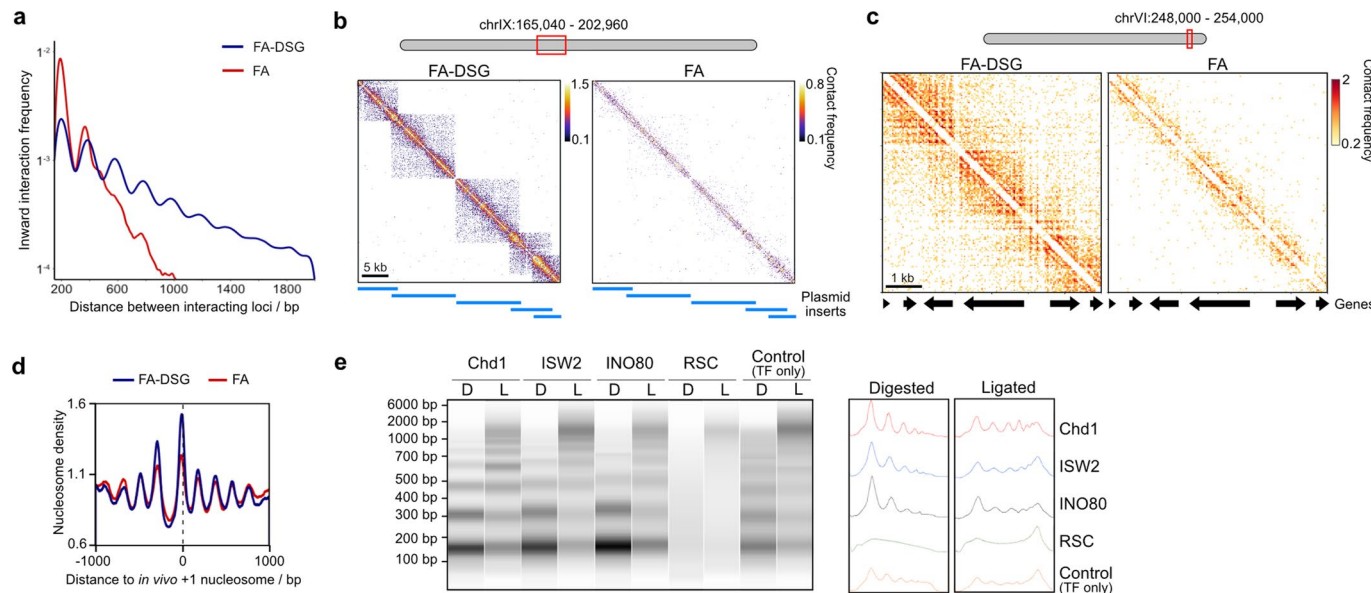

**Extended Data Fig. 3 | Establishment of the *in vitro* Micro-C procedure.**
**a**, Distance decay curve of inward-facing nucleosome interactions for indicated crosslinking conditions of *in vitro* Micro-C of SGD chromatin generated in the presence of INO80, Abf1 and Reb1. 0.05% formaldehyde (FA) and 0.75 mM disuccinimidyl glutarate (DSG) was used. **b**, Contact matrices of a 37.92 kbp region of *in vitro* Micro-C from chromatin samples described in panel **a. c**,

Contact matrices of a 6 kbp region of *in vitro* Micro-C from chromatin samples as described in panel **a. d**, Nucleosome occupancy profiles of *in vitro*-reconstituted chromatin from chromatin samples as described in panel **a. e**, Capillary gel electrophoresis of *in vitro* Micro-C samples after MNase-digestion (D) and after proximity ligation (L). Chromatin was incubated with the indicated remodeler and the TFs Abf1 and Reb1 or with the TFs only.

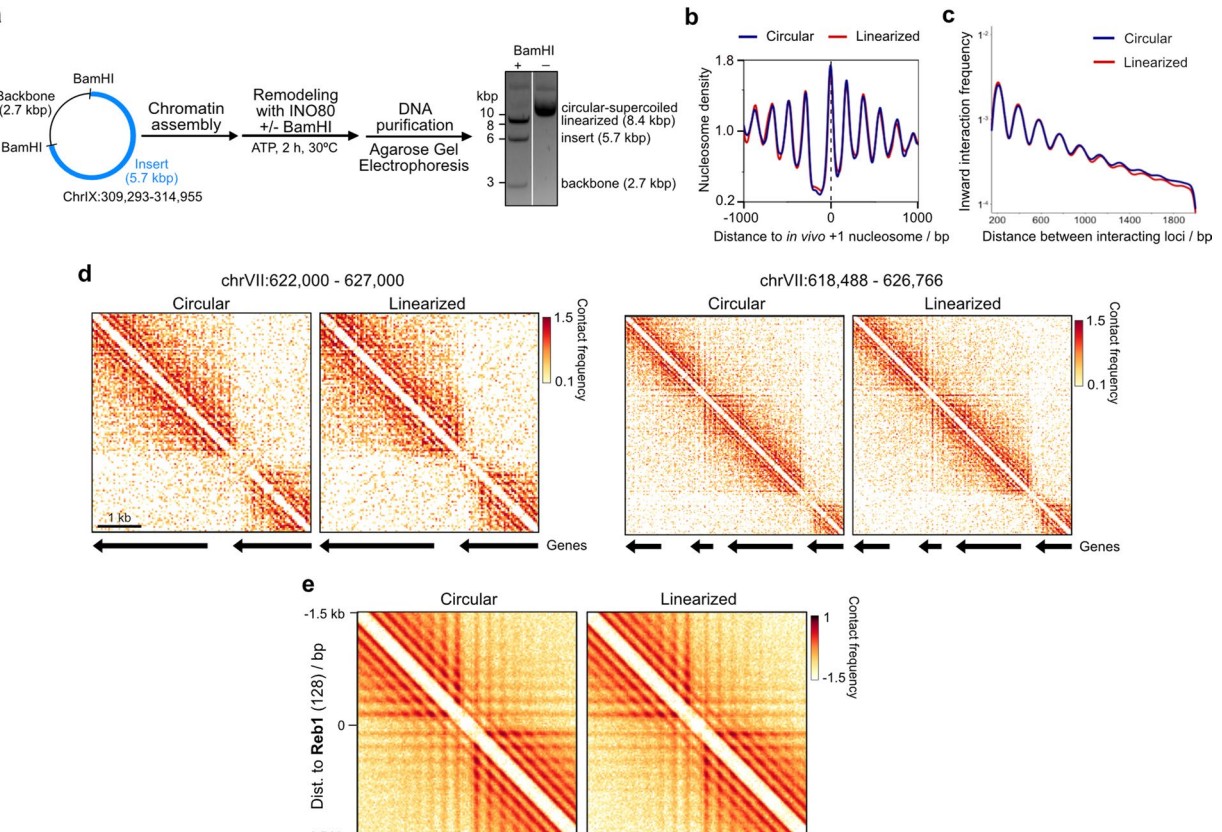

**Extended Data Fig. 4 | Comparison of circular and linearized *in vitro* chromatin. a**, Assessment of BamHI digestion efficiency during chromatin remodeling. An individual, chromatinized plasmid with two BamHI cut sites was incubated with INO80, Abf1, Reb1, and with or without BamHI. Analysis of the fragment sizes with agarose gel electrophoresis shows that the digestion of the plasmid into two separate fragments is not complete, but that the vast majority of the chromatinized plasmid has been cut at least once and has therefore been linearized. **b**, Nucleosome occupancy profiles of *in vitro*-reconstituted chromatin incubated with INO80, Abf1 and Reb1 alone (circular) or additionally with the restriction enzyme BamHI (linearized). **c**, Distance decay curve of inward-facing nucleosome interactions of *in vitro* Micro-C from chromatin samples as described in panel **b**. **d**, Contact matrices displaying in vitro Micro-C data for a ~5 and ~8 kbp region of reconstituted chromatin as described in panel **b**. **e**, Pile-up analysis of contact matrices from chromatin samples as described in panel **b** aligned at Reb1 binding sites.

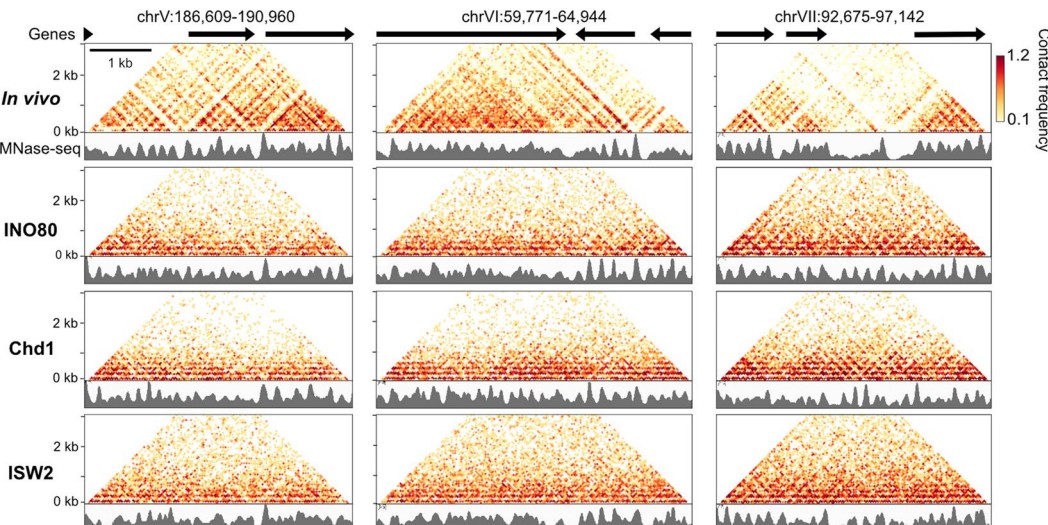

**Extended Data Fig. 5 | *In vitro* chromatin conformations in regions without Abf1 and Reb1 binding sites.** Contact matrices displaying *in vivo*[13] and *in vitro* Micro-C data and corresponding nucleosome occupancy profiles (MNase-seq) for three genomic regions with low Abf1 and Reb1 occupancy. Gene annotation is shown at the top[19]. Chromatin used for *in vitro* Micro-C was incubated with the indicated remodeler and the TFs Abf1 and Reb1. Micro-C data are plotted as log$_{10}$ interaction counts at 40 bp resolution.

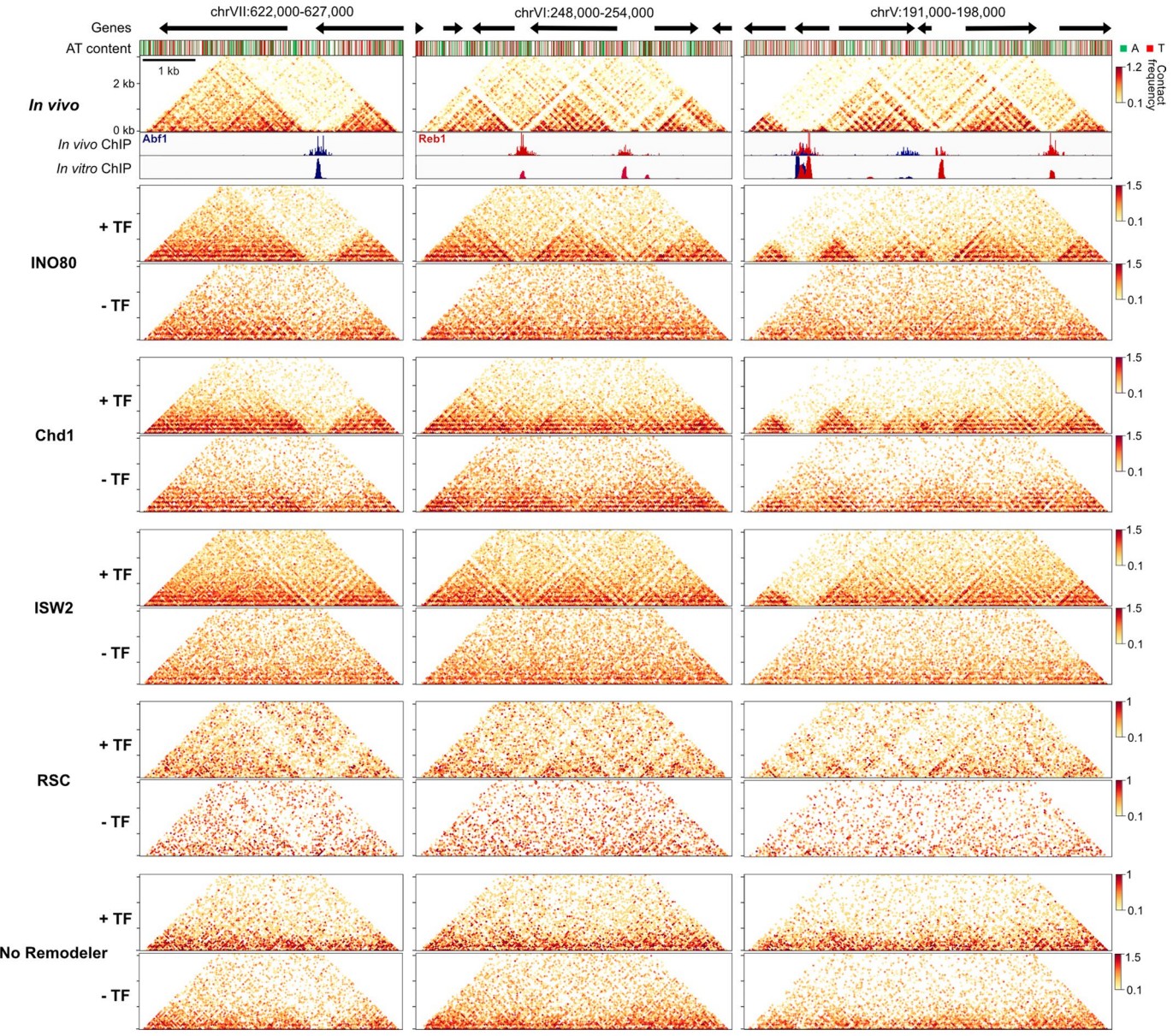

**Extended Data Fig. 6 | Comparison of *in vitro* chromatin structure in presence and absence of chromatin remodelers and transcription factors.** Contact matrices displaying *in vivo*[13] and *in vitro* Micro-C data for three genomic regions. Gene and sequence composition annotation are shown at the top (A = adenine; T = thymine). *In vivo*[19] and *in vitro* ChIP-seq data for Abf1 (blue) and Reb1 (red) are shown below the *in vivo* data. Chromatin used for *in vitro* Micro-C was incubated with or without chromatin remodelers and the TFs Abf1 and Reb1 as indicated. Micro-C data are plotted as $\log_{10}$ interaction counts at 40 bp resolution.

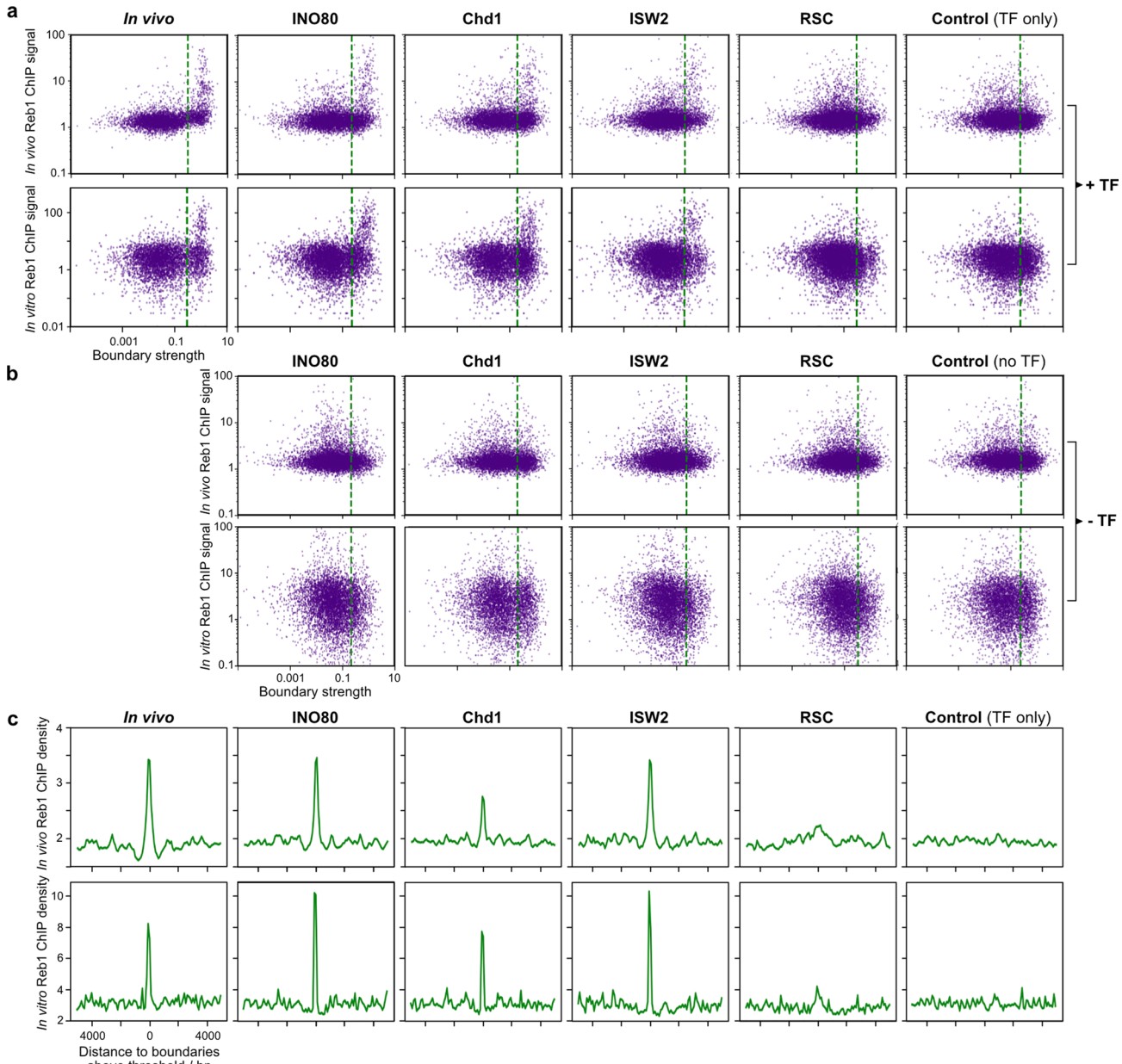

**Extended Data Fig. 7 | Correlation between boundary strength and transcription factor enrichment. a**, TF enrichment derived from *in vivo*[79] (top) and *in vitro* (bottom) ChIP-seq data as a function of boundary strength defined by insulation scores plotted with 800 bp sliding windows derived from *in vivo*[13] and *in vitro* Micro-C data. Chromatin used for *in vitro* Micro-C was incubated with the indicated remodeler and the TFs Abf1 and Reb1 or with the TFs only. Green dashed lines denote strong boundaries based on Li automated thresholding criteria[76]. **b**, Analysis as described in panel **a** for chromatin incubated with the indicated remodeler but without TFs. **c**, *In vivo*[79] (top) and *in vitro* (bottom) ChIP-seq signal aligned at and averaged over strong, thresholded boundary sites derived from panel **a**.

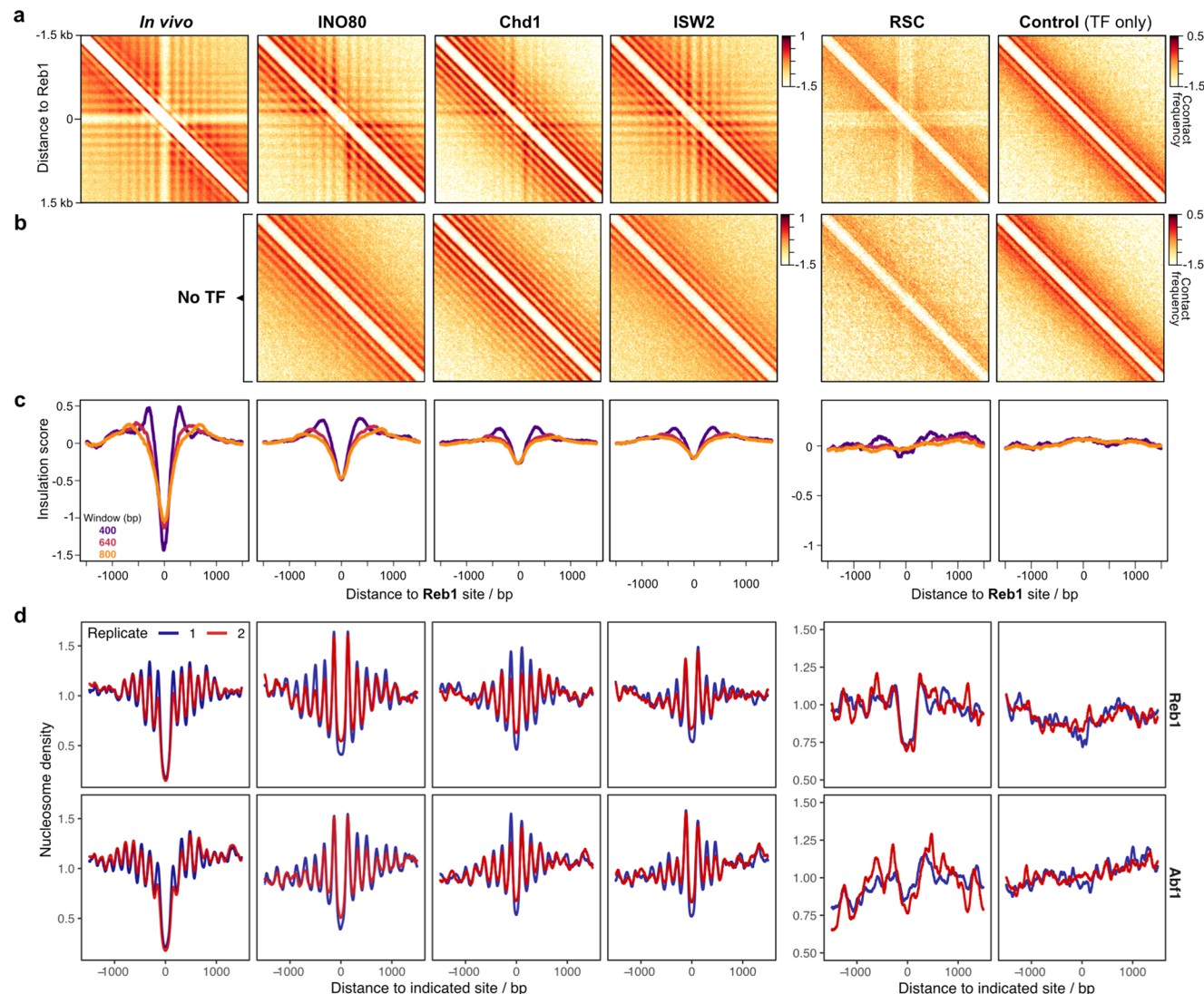

**Extended Data Fig. 8 | Correlation between insulation strength and nucleosome positioning. a,** Pile-up analysis of contact matrices aligned at Reb1 binding sites. Chromatin was incubated with the indicated remodeler and the TFs Abf1 and Reb1 or with the TFs only. *In vivo* Micro-C data[13] are shown for comparison. $\log_{10}$ interaction counts are plotted at 20 bp resolution. **b,** Pile-up analysis of contact matrices as described in panel **a**. Chromatin was incubated with the indicated remodeler but without the TFs. **c,** Insulation scores derived from Micro-C data shown in panel **a** calculated at 80 bp resolution. Three different sliding windows are shown. **d,** Nucleosome occupancy profiles (MNase-seq) of individual replicates aligned at Abf1 or Reb1 binding sites corresponding to panels **a** and **c**.

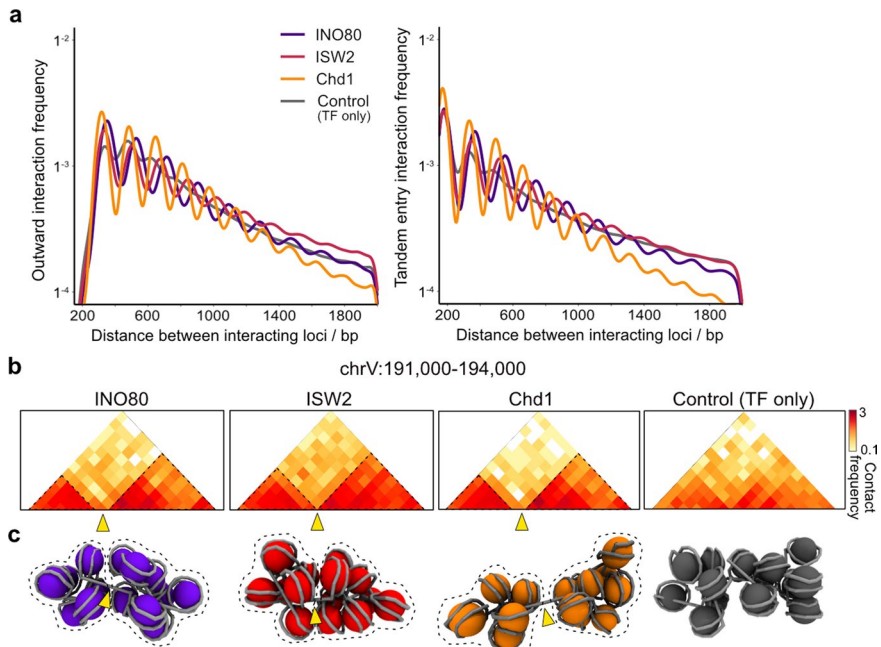

**Extended Data Fig. 9 | Chromatin domain compaction increases with longer linker lengths. a**, Interaction frequency as a function of genomic distance plotted for outward and tandem interactions of nucleosomes as derived from *in vitro* Micro-C data from chromatin incubated with the indicated remodeler and the TFs Abf1 and Reb1 or with the TFs only. $\log_{10}$ interaction frequencies are plotted. **b**, Nucleosome-binned contact matrices displaying *in vitro* Micro-C data for the indicated region. **c**, Molecular dynamics simulations[33] of regions shown in panel **b**. Arrowheads point toward boundaries at NFRs corresponding to panel **b**. Stippled lines highlight chromatin domains corresponding to panel **b**.

# Reporting Summary

## Statistics

For all statistical analyses, confirm that the following items are present in the figure legend, table legend, main text, or Methods section.

| n/a | Confirmed | |
|---|---|---|
| ☐ | ☒ | The exact sample size (*n*) for each experimental group/condition, given as a discrete number and unit of measurement |
| ☐ | ☒ | A statement on whether measurements were taken from distinct samples or whether the same sample was measured repeatedly |
| ☐ | ☒ | The statistical test(s) used AND whether they are one- or two-sided<br>*Only common tests should be described solely by name; describe more complex techniques in the Methods section.* |
| ☒ | ☐ | A description of all covariates tested |
| ☒ | ☐ | A description of any assumptions or corrections, such as tests of normality and adjustment for multiple comparisons |
| ☐ | ☒ | A full description of the statistical parameters including central tendency (e.g. means) or other basic estimates (e.g. regression coefficient) AND variation (e.g. standard deviation) or associated estimates of uncertainty (e.g. confidence intervals) |
| ☐ | ☒ | For null hypothesis testing, the test statistic (e.g. *F*, *t*, *r*) with confidence intervals, effect sizes, degrees of freedom and *P* value noted<br>*Give P values as exact values whenever suitable.* |
| ☒ | ☐ | For Bayesian analysis, information on the choice of priors and Markov chain Monte Carlo settings |
| ☒ | ☐ | For hierarchical and complex designs, identification of the appropriate level for tests and full reporting of outcomes |
| ☐ | ☒ | Estimates of effect sizes (e.g. Cohen's *d*, Pearson's *r*), indicating how they were calculated |

*Our web collection on statistics for biologists contains articles on many of the points above.*

## Software and code

Policy information about availability of computer code

| Data collection | Illumina NextSeq 550. |
|---|---|
| Data analysis | Trim Galore v.0.3.1; FLASH v.1.2.11; BLAT v.35; Bowtie v.1.2.1.1; Bowtie2 v.2.3.5; GenomicAlignments v.1.30.0; igv v.2.8.6; MEME 5.5.2; MACS2 2.2.8; HiC-Pro v.2.11.1; cooler v.0.8.11; cooltools v.0.5.1; hiCExplorer v.3.6; MCC pipeline v.1 (https://github.com/jojdavies/Micro-Capture-C); R Studio (v.2023.030+386; R v.4.1.3); see Methods section for full details. |

For manuscripts utilizing custom algorithms or software that are central to the research but not yet described in published literature, software must be made available to editors and reviewers. We strongly encourage code deposition in a community repository (e.g. GitHub). See the Nature Portfolio guidelines for submitting code & software for further information.

## Data

Policy information about availability of data

All manuscripts must include a data availability statement. This statement should provide the following information, where applicable:

- Accession codes, unique identifiers, or web links for publicly available datasets
- A description of any restrictions on data availability
- For clinical datasets or third party data, please ensure that the statement adheres to our policy

All raw sequencing data and processed data generated in this study are available for download at http://www.ncbi.nlm.nih.gov/geo/ via GEO accession number

## Human research participants

Policy information about studies involving human research participants and Sex and Gender in Research.

| | |
|---|---|
| Reporting on sex and gender | n/a |
| Population characteristics | n/a |
| Recruitment | n/a |
| Ethics oversight | n/a |

Note that full information on the approval of the study protocol must also be provided in the manuscript.

# Field-specific reporting

Please select the one below that is the best fit for your research. If you are not sure, read the appropriate sections before making your selection.

☒ Life sciences          ☐ Behavioural & social sciences          ☐ Ecological, evolutionary & environmental sciences

For a reference copy of the document with all sections, see nature.com/documents/nr-reporting-summary-flat.pdf

# Life sciences study design

All studies must disclose on these points even when the disclosure is negative.

| | |
|---|---|
| Sample size | No statistical methods were used to pre-determine sample sizes but our sample sizes are similar to those reported in previous publications. The data presented in the manuscript represent the averages of multiple replicates as stated in each of the figures and described in detail in the Methods section. These sample sizes were chosen to generate data at sufficient depth and assess differences between conditions robustly. These sample sizes are sufficient, since the observed effects of interest are clearly detectable between conditions and robust across replicates. |
| Data exclusions | No data were excluded. |
| Replication | Experiments were performed independently at least two times as described in detail in the Methods section and all attempts were successful. |
| Randomization | Since the researchers need to know the experimental condition in order to perform the experiments successfully, randomization is not relevant for our study. |
| Blinding | All samples were analyzed with the same pipeline, in which interactions are detected by scripts without interference of the researchers. Since potential expectations of the researchers cannot influence the data analysis and results, blinding is not relevant to this study. |

# Reporting for specific materials, systems and methods

We require information from authors about some types of materials, experimental systems and methods used in many studies. Here, indicate whether each material, system or method listed is relevant to your study. If you are not sure if a list item applies to your research, read the appropriate section before selecting a response.

## Materials & experimental systems

| n/a | Involved in the study |
|---|---|
| ☐ | ☒ Antibodies |
| ☐ | ☒ Eukaryotic cell lines |
| ☒ | ☐ Palaeontology and archaeology |
| ☒ | ☐ Animals and other organisms |
| ☒ | ☐ Clinical data |
| ☒ | ☐ Dual use research of concern |

## Methods

| n/a | Involved in the study |
|---|---|
| ☐ | ☒ ChIP-seq |
| ☒ | ☐ Flow cytometry |
| ☒ | ☐ MRI-based neuroimaging |

# Antibodies

| | |
|---|---|
| Antibodies used | Anti-FLAG M2 Affinity Gel (A2220, Merck) was used for protein purification. |
| Validation | Anti-FLAG M2 Affinity Gel (A2220, Merck) was used for protein purification and thus required no additional validation. |

# Eukaryotic cell lines

Policy information about cell lines and Sex and Gender in Research

| | |
|---|---|
| Cell line source(s) | Hi5 cells: Expression Systems (#94-002F), Tni insect cells in ESF921 media.<br>Saccharomyces cerevisiae: RSC2-TAP-HIS3 (YSC1177-YLR357W), Dharmacon, TAP-tagged open reading frame library.<br>Saccharomyces cerevisiae: INO80 overexpression strain (yAE86), obtained from Kurat et al. Molecular Cell 2017). |
| Authentication | Authentication of the yeast strains was performed by PCR. The insect cells were authenticated by the manufacturer. |
| Mycoplasma contamination | The insect and yeast cell lines used for protein expression were not tested for mycoplasma contamination. |
| Commonly misidentified lines<br>(See ICLAC register) | None. |

# ChIP-seq

## Data deposition

☒ Confirm that both raw and final processed data have been deposited in a public database such as GEO.

☒ Confirm that you have deposited or provided access to graph files (e.g. BED files) for the called peaks.

| | |
|---|---|
| Data access links<br>*May remain private before publication.* | http://www.ncbi.nlm.nih.gov/geo/ via GEO accession number GSE220647 (token: wnyncsemtjybfsp) |
| Files in database submission | Fastq files and bigwig files for the following samples were uploaded to GEO:<br><br>Replicate1_ control_IP_Reb1<br>Replicate1_TFs-only_IP_Reb1<br>Replicate1_INO80_IP_Reb1<br>Replicate1_ISW2_IP_Reb1<br>Replicate1_Chd1_IP_Reb1<br>Replicate1_RSC_IP_Reb1<br>Replicate2_ control_IP_Reb1<br>Replicate2_TFs-only_IP_Reb1<br>Replicate2_INO80_IP_Reb1<br>Replicate2_ISW2_IP_Reb1<br>Replicate2_Chd1_IP_Reb1<br>Replicate2_RSC_IP_Reb1<br>Replicate1_ control_IP_Abf1<br>Replicate1_TFs-only_IP_Abf1<br>Replicate1_INO80_IP_Abf1<br>Replicate1_ISW2_IP_Abf1<br>Replicate1_Chd1_IP_Abf1<br>Replicate1_RSC_IP_Abf1<br>Replicate2_ control_IP_Abf1<br>Replicate2_TFs-only_IP_Abf1<br>Replicate2_INO80_IP_Abf1<br>Replicate2_ISW2_IP_Abf1<br>Replicate2_Chd1_IP_Abf1<br>Replicate2_RSC_IP_Abf1<br>Replicate1_ control_Input_Reb1<br>Replicate1_TFs-only_Input_Reb1<br>Replicate1_INO80_Input_Reb1<br>Replicate1_ISW2_Input_Reb1<br>Replicate1_Chd1_Input_Reb1<br>Replicate1_RSC_Input_Reb1<br>Replicate2_ control_Input_Reb1<br>Replicate2_TFs-only_Input_Reb1<br>Replicate2_INO80_Input_Reb1<br>Replicate2_ISW2_Input_Reb1<br>Replicate2_Chd1_Input_Reb1<br>Replicate2_RSC_Input_Reb1<br>Replicate1_ control_Input_Abf1<br>Replicate1_TFs-only_Input_Abf1 |

Replicate1_INO80_Input_Abf1
Replicate1_ISW2_Input_Abf1
Replicate1_Chd1_Input_Abf1
Replicate1_RSC_Input_Abf1
Replicate2_ control_Input_Abf1
Replicate2_TFs-only_Input_Abf1
Replicate2_INO80_Input_Abf1
Replicate2_ISW2_Input_Abf1
Replicate2_Chd1_Input_Abf1
Replicate2_RSC_Input_Abf1

Genome browser session
(e.g. UCSC)

https://tinyurl.com/yurw2hwf

## Methodology

Replicates

Two biological replicates for each experimental condition were performed.

Sequencing depth

The samples were sequenced using the Illumina NextSeq550 sequencer in 42 bp paired-end mode to a sequencing depth of ~5 Mio. reads per sample.

Antibodies

Strep-Tactin® (IBA, 2-1613-002). Please note that Strep-Tactin® (modified Streptavidin) was used instead of an Antibody.

Peak calling parameters

MACS2 (default parameters).

Data quality

The quality of the in vitro Reb1 and Abf1 ChIP-seq data was assessed by comparing peaks and discovered PWM motifs to previously published in vivo Reb1 and Abf1 ChIP-seq data sets and motifs (Gutin et al. 2018, Cell Reports; Rossi et al. 2021, Nature).

Software

Reads were mapped to the SacCer3 genome (R64-1-1 assembly) using Bowtie, omitting multiple matches. Peak calling was performed with MACS2 and motif discovery in the regions of the called peaks was performed using MEME.

