## [Peer Review File · Nature Genetics]

Peer Review Information

Manuscript Title: In vitro reconstitution of chromatin domains reveals a role for nucleosome positioning in 3D genome organization

Corresponding author name(s): Dr Marieke Oudelaar, Dr Elisa Oberbeckmann

Reviewer Comments & Decisions:

Decision Letter, initial version:

9th May 2023

Dear Dr Oudelaar,

hope this email finds you well.

Your Article entitled "In vitro reconstitution of chromatin domains" has now been seen by 3 referees, whose comments are attached. While they find your work of potential interest, they have raised serious concerns which in our view are sufficiently important that they preclude publication of the work in Nature Genetics, at least in its present form.

The most concerning comment raised by all three reviewers is that your findings are not generalizable to the situation in vivo (where usually genomic interactions occur at higher scale), as you are assessing a synthetic DNA construct that is only 10 kb long. While Reviewer #1 would be fine with having this as a textual caveat for your work, in contrast, Reviewer #2 thinks it makes your study too preliminary for publication on Nature Genetics. Reviewer #3 falls somewhere in between, suggesting this is of potentially broad interest, but with multiple concerns on the strength of evidence as yet provided; they do, however, propose further experiments to address these issues.

In our reading, it seems that Referee #2 requires an application of your in vitro system to mammalian-scale DNA; we would, of course, strongly welcome this but acknowledge this is likely a difficult request to fulfil. However, given the strong support of Reviewer #1, and the possibility that Referee #3 will be supportive if you are able to fully address their experimental comments, we concluded that there is a potential path to publication.

Should further experimental data allow you to fully address these criticisms we would be willing to consider an appeal of our decision (unless, of course, something similar has by then been accepted at Nature Genetics or appeared elsewhere). This includes submission or publication of a portion of this work someplace else.

We hope you understand that until we have read the revised manuscript in its entirety we cannot promise that it will be sent back for peer review.

If you are interested in attempting to revise this manuscript for submission to Nature Genetics in the future, please contact me to discuss a potential appeal. Otherwise, we hope that you find our referees' comments helpful when preparing your manuscript for resubmission elsewhere.

Best wishes,
Chiara

Chiara Anania, PhD
Associate Editor
Nature Genetics
<https://orcid.org/0000-0003-1549-4157>

Referee expertise:

Referee #1: chromosome biology - chromatin - nuclear architecture

Referee #2: genome organization - gene expression regulation

Referee #3: epigenetic regulation and chromatin architecture

Reviewers' Comments:

Reviewer #1:

Remarks to the Author:

Quilian et al. have reconstituted chromatin domains in vitro. To do so, they use a plasmid library of ~10kb inserts representing sections of chromosomes V-IX and they assemble high-density nucleosomes on them. Addition of chromatin remodelers and transcription factors leads to domain formation similar to in vivo domains as assessed by micro-C analysis. Formation of domain boundaries required regular nucleosome positioning and they find correlation between boundary strength and width of nucleosome free regions. Longer nucleosome linker length correlated with chromatin domain compaction. Finally, computational modeling recreates some of the domain features.

This is a very timely and innovative study. Most of what we think we know about chromatin organization comes from correlations or from loss of function approaches in which proteins are knocked down or disruptive mutations introduced. This study uses a novel bottom-up synthetic biology approach to re-create chromatin structures.

The authors make several important findings including the demonstration that in yeast the presence of transcription factors and chromatin remodelers is sufficient to generate chromatin domains and no cohesin or CTCF is required to define chromatin domains and they find that domain boundaries are defined by nucleosome free regions. The finding that distinct remodelers give rise to distinct chromatin features is interesting as well and maybe a little under-emphasized by the authors. Overall, this is an

important novel approach to studying chromatin and they are important findings. They contribute significantly to our understanding of how chromatin domains form.

A limitation of the study is that the domains created are small (~10kb), especially relative to those in higher eukaryotes. It is not clear that these findings apply to larger TADs. This is not a fatal flaw, but the authors should make it very clear that they do not mean to generalize these findings to other species. Some statements regarding universal applicability of the finding should be toned down, for example, the end of the first paragraph of the discussion or in the abstract.

Reviewer #2:

Remarks to the Author:

This paper by Quillian et al. takes a bottom-up approach to understanding chromatin by using in vitro reconstitution to generate chromatin from plasmids containing 10 kb segments of yeast chromosomes. They develop an MNase based 3C method to map local architecture and nucleosome organization and then study how different chromatin proteins – including INO80, Chd1, ISW2, and RSC – regulate the in vitro reconstitution and short-range interactions (a few kb). They find that NFRs correlate with the boundary between tiny “chromatin domains” and speculate on the relevance to human 3D chromatin architecture.

I like bottom-up approaches and bottom-up reconstitution of human-relevant 3D architecture would be a major achievement. Unfortunately, the 10kb plasmid system used here is not particularly relevant to yeast 3D architecture and much less human 3D architecture where interactions tend to require hundreds of kb. As such this reads like a preliminary proof-of-concept using an unfortunate plasmid system. While the data and analysis seem solid and while the approach could be interesting, I unfortunately do not think the present paper is particularly groundbreaking and certainly not at the level one would typically expect for a Nature Genetics paper.

COMMENTS

PLASMIDS: It is puzzling to me that the authors would choose to use plasmids containing 10kb of yeast DNA. First, the 3D structure of a ~10-20kb plasmid clearly has no relevance to a large long linear chromosome (I know they sometimes linearize the plasmid, but this is still not relevant and the maps look different from circular plasmids). Second, many 3D interactions tend to occur across scales including tens to hundreds of kb, even in yeast. But using tiny DNA segments, they make it impossible to observe the interactions that make 3D architecture interesting.

CHROMATIN DOMAINS: The authors repeatedly refer to “chromatin domains” without defining this term. In the 3D architecture field, there is already too many terms like TADs, insulated neighborhood, loop domain, contact domain, compartment, hub, tether, etc. It is not clear what a chromatin domain is. Please explicitly define it and say if/how this is relevant to other common terms like TAD, compartment, or loop domain.

HUMAN 3D ARCHITECTURE: The authors speculate on relevance to human 3D architecture and include the sentence “This demonstrates that neither loop extrusion nor transcription are required for domain formation.”. Relevant to the above this is quite misleading and confusing in my opinion – the domains, TADs measuring hundreds of kb, that are thought to form through loop extrusion clearly have no

relevant relationship with 10kb plasmids and it seems misleading to indicate so.

Reviewer #3:

Remarks to the Author:

I have reviewed this manuscript previously for a different journal, and the same comments are shared here, as no relevant changes were made to address the previously shared concerns.

In the manuscript titled "In vitro reconstruction of chromatin domains", Quililan, Oberbeckmann et al. investigate the mechanisms important for the folding of chromatin in *S. cerevisiae*. The authors developed a strategy to reconstitute the chromatin in vitro and adapted the MNase-based 3C protocol to map its 3D conformation. By incubating plasmids that contain parts of the yeast genome with different purified chromatin remodelers (i.e. INO80, ISW2, Chd1) and general transcription factors (i.e. Abf1 and Reb1) they show that the regular spaced and phased nucleosomes are sufficient for the formation of chromatin domains, and that these are similar to those found in vivo. Furthermore, they show that the compaction of the domains depends on the size of the nucleosome linker which varied with the chromatin remodeller.

The line of research pioneered in this manuscript is highly promising for the development of its field, but its strong claims are currently not convincingly supported. The following major comments must be addressed before publication.

Major comments:

- It has been previously shown that binding of multiple transcription factors can drive insulation. The authors make the assumption that Abf1 and Reb1 bind in vitro based on the in vivo ChIP-seq data, but the manuscript does not directly demonstrate that Abf1 and Reb1 TFs are actually able to bind DNA in the in vitro conditions chosen, in the absence of chromatin remodellers. To address this, ChIP-seq experiments are necessary for both Abf1 and Reb1 in the different in vitro conditions tested (in presence of INO80, ISW2, Chd1 and general transcription factors). Furthermore, to strengthen the claim that nucleosome spacing is the main driver of chromatin domain formation, it is also necessary that assays without the transcription factors are performed and the results included in the manuscript as controls. The correlations between TF factor enrichment and the boundary strengths shown in Extended Data Fig 4a also need to be calculated based on the actual experimental in vitro ChIP-seq data that is currently missing.

- The main claim of the manuscript is that: "the formation of regular and phase nucleosome arrays is sufficient to reconstitute native chromatin domains" and it is concluded that transcription is not essential for the formation of chromatin domains in yeast. Nonetheless, many biochemical processes such as transcription, DNA repair etc., have an impact on the regular nucleosome positioning. Binding of the transcription machinery and transcription itself have been previously shown to impact chromatin condensation status and implicitly on nucleosome spacing. The authors focused their analysis on specific loci, enriched for Abf1 and Reb1 binding motifs, and did not address directly the function of transcription or other cellular biochemical processes on nucleosome spacing. Furthermore, there is evidence from work on mammalian chromatin, that transcription is important for boundary formation in some cases and not essential in others. To understand whether transcription in yeast might play a

similar role the authors would need to look at other genomic areas and/or devise in vitro experiments that address this matter directly by for example studying chromatin configuration with and without the transcription machinery. Therefore, it is essential that broad claims about transcription not being important for domain formation are removed or correctly rephrased to be clearly made specific to the in vitro system used, without unsupported generalisation.

- The manuscript includes examples of regions that are enriched for the transcription factors investigated. To address the role of transcription factors in shaping chromatin structure, it is important to also investigate regions that are not enriched for these particular transcription factors. How does the conformation of domains not bound by the transcription factors investigated here compare to in vivo? If such investigations cannot be made, then the conclusions and interpretations must be down-toned to report on the specific system that was used, acknowledging that a lot more work is necessary before generalisations can be made.

Other comments

- It is essential to explicitly clarify whether circular or linearised Micro-C data were used for most of the analysis. The matrices shown on Fig1,d suggest that regions surrounding BamHI restriction sites still form chromatin contacts after the digestion. It is important to acknowledge this observation in the manuscript and comment on whether there is incomplete DNA digestion in the sample or another problem.

- To help understand how much can the work be generalised, the manuscript should provide further information about the region selected for Fig 2, namely its gene density, GC content, etc relative to the *S. cerevisiae* genome, to help understand to what extent the region is somehow representative.

- Several studies have highlighted differences in chromatin organisation between ligation- and microscopy-based technologies, and suggest a more cautious interpretation of the results (for example Williamson et al., 2014; DOI: 10.1101/gad.251694.114). The claims of the current manuscript are strong and should acknowledge that the results might be inherently biased by the cross-linking events. For instance, ligation-based technologies have a strong bias in capturing interactions that are strongly bound by proteins. It is therefore possible that the strength of the domains observed here is biased by the number of proteins cross-linked, and it cannot be excluded that the domains form and/or are weaker in the absence of TF binding.

- It is recommended that the authors' included a statement about whether the proposed mechanisms are limited to 10kb chromatin domains or could potentially contribute to the formation of larger-scale chromatin domains such as mammalian TADs.

Decision Letter, Appeal:

17th May 2023

Dear Marieke,

Thank you again for your email asking us to reconsider our decision on your manuscript "In vitro reconstitution of chromatin domains". I have now discussed the points of your appeal with my

colleagues, and we think that you have some valid points. We therefore invite you to revise your manuscript along the lines that you propose. We will be then happy to send the revised documents back to reviewers. We appreciate that you included an estimation of the time you will need to revise your manuscript and we think 3-4 months are also suitable for us.

When preparing a revision, please ensure that it fully complies with our editorial requirements for format and style; details can be found in the Guide to Authors on our website (<http://www.nature.com/ng/>).

Please be sure that your manuscript is accompanied by a separate letter detailing the changes you have made and your response to the points raised. At this stage we will need you to upload:

1) a copy of the manuscript in MS Word .docx format.

2) The Editorial Policy Checklist:

<https://www.nature.com/documents/nr-editorial-policy-checklist.pdf>

3) The Reporting Summary:

(Here you can read about the role of the Reporting Summary in reproducible science:

<https://www.nature.com/news/announcement-towards-greater-reproducibility-for-life-sciences-research-in-nature-1.22062>)

Please use the link below to be taken directly to the site and view and revise your manuscript:

[redacted]

If you have any questions, do not hesitate to contact me by email. We can also schedule a call if you prefer.

Thank you.

With kind wishes,
Chiara

Chiara Anania, PhD
Associate Editor
Nature Genetics
<https://orcid.org/0000-0003-1549-4157>

Author Rebuttal to Initial comments

Reviewer #1:

Remarks to the Author:

Quilian et al. have reconstituted chromatin domains in vitro. To do so, they use a plasmid library

of ~10kb inserts representing sections of chromosomes V-IX and they assemble high- density nucleosomes on them. Addition of chromatin remodelers and transcription factors leads to domain formation similar to in vivo domains as assessed by micro-C analysis. Formation of domain boundaries required regular nucleosome positioning and they find correlation between boundary strength and width of nucleosome free regions. Longer nucleosome linker length correlated with chromatin domain compaction. Finally, computational modeling recreates some of the domain features.

This is a very timely and innovative study. Most of what we think we know about chromatin organization comes from correlations or from loss of function approaches in which proteins are knocked down or disruptive mutations introduced. This study uses a novel bottom-up synthetic biology approach to re-create chromatin structures.

The authors make several important findings including the demonstration that in yeast the presence of transcription factors and chromatin remodelers is sufficient to generate chromatin domains and no cohesin or CTCF is required to define chromatin domains and they find that domain boundaries are defined by nucleosome free regions. The finding that distinct remodelers give rise to distinct chromatin features is interesting as well and maybe a little under-emphasized by the authors. Overall, this is an important novel approach to studying chromatin and they are important findings. They contribute significantly to our understanding of how chromatin domains form.

A limitation of the study is that the domains created are small (~10kb), especially relative to those in higher eukaryotes. It is not clear that these findings apply to larger TADs. This is not a fatal flaw, but the authors should make it very clear that they do not mean to generalize these findings to other species. Some statements regarding universal applicability of the finding should be toned down, for example, the end of the first paragraph of the discussion or in the abstract.

We would like to thank the Reviewer for their positive evaluation of our work. We agree with the Reviewer that it is important to highlight that we have studied yeast chromatin, in which domains are relatively small, and that our findings are not directly generalizable to higher eukaryotes. We have made changes throughout the text to make this clear. In the Abstract and in the first paragraph of the Discussion, we have clarified that our findings apply to yeast. In the sections in the Discussion in which we discuss potential implications of our study for higher eukaryotes, we have made it clear that these are speculative. This includes our discussion of the role of transcription (Lines 468-478) and loop extrusion (Lines 480-489) in the formation of chromatin domains. In addition, we have added a paragraph to the Discussion to clarify the limitations of our

study (Lines 424-442). We have copied the key sections of these paragraphs below for convenience.

Lines 468-478:

Although the relatively simple organization of the yeast genome is not directly representative of the more complex organization of mammalian genomes, it is interesting to point out that transcription and nucleosome positioning are similarly intertwined in mammals. We therefore speculate that it is possible that observations that have been interpreted as a driving role for transcription in genome folding in mammals could also reflect associated patterns of nucleosome positioning. Examples include the enrichment of active promoters at domain boundaries^{61,62}, which are also characterized by strong NFRs, as well as the interpretation of transcription perturbation experiments, which are generally also associated with changes in nucleosome positioning⁵². Although speculative, our findings therefore suggest that the role of transcription in genome organization in vivo may, to some extent, be mediated by the influence of transcription on nucleosome positioning.

Lines 480-489:

Since we were able to reconstitute in-vivo-like chromatin domains in absence of SMC proteins, our experiments also indicate that loop extrusion is not per se required for basic domain organization of the yeast genome in interphase. It should be noted though that cohesin-dependent loop extrusion is thought to play an important role in the regulation of yeast chromatin structure during S-phase and mitosis⁶³⁻⁶⁶. However, the notion that loop extrusion may not be required for the organization of the yeast genome during interphase is consistent with low levels of cohesin occupancy on chromatin during interphase⁶⁷ and with cohesin perturbation studies in yeast⁶³⁻⁶⁶. Together, this indicates that the chromatin domains that form in yeast interphase are not comparable to vertebrate TADs, which form via a loop extrusion process mediated by SMC proteins^{68,69}, and may be more similar to compartments.

Lines 424-442:

The SGD reconstitution system that we have used in this study has the advantage that it allows for reconstitution of chromatin with in-vivo-like features using a native, chromosome-wide DNA template²³. However, this system also has limitations. Most importantly, the reconstituted chromatin fibers are restricted in size, as they are generated from a genomic plasmid library with ~10 kbp inserts (Fig. 1a). This system is therefore limited to analyzing features of chromatin organization on a scale of ~10 kbp and cannot be used (in its current form) to analyze larger-scale features of genome folding. However, since the yeast genome is characterized by short-range interactions and folds into relatively small (~5 kbp) domains in interphase¹⁹, the SGD reconstitution system is suitable to study the formation of chromatin domains and the features of

their boundaries in yeast. Furthermore, since Abf1 and Reb1 are the only transcription factors that are present in the reconstitution reactions, this system is limited to analyzing regions that are bound by these transcription factors and not informative for in vitro analysis of regions that are bound by other transcription factors in vivo. Another limitation of our study is that our analyses of the organization of reconstituted chromatin are solely based on 3C data. 3C is based on chromatin crosslinking, digestion, and proximity ligation, which is reflected in the resulting interaction patterns. These patterns may therefore be biased towards detecting interactions between regions that are strongly bound by proteins⁵¹. It would therefore be of interest to combine future reconstitution studies with microscopy-based approaches to measure the organization and dynamics of chromatin.

Reviewer #2:

Remarks to the Author:

This paper by Quillian et al. takes a bottom-up approach to understanding chromatin by using in vitro reconstitution to generate chromatin from plasmids containing 10 kb segments of yeast chromosomes. They develop an MNase based 3C method to map local architecture and nucleosome organization and then study how different chromatin proteins – including INO80, Chd1, ISW2, and RSC – regulate the in vitro reconstitution and short-range interactions (a few kb). They find that NFRs correlate with the boundary between tiny “chromatin domains” and speculate on the relevance to human 3D chromatin architecture.

I like bottom-up approaches and bottom-up reconstitution of human-relevant 3D architecture would be a major achievement. Unfortunately, the 10kb plasmid system used here is not particularly relevant to yeast 3D architecture and much less human 3D architecture where interactions tend to require hundreds of kb. As such this reads like a preliminary proof-of-concept using an unfortunate plasmid system. While the data and analysis seem solid and while the approach could be interesting, I unfortunately do not think the present paper is particularly groundbreaking and certainly not at the level one would typically expect for a Nature Genetics paper.

We would like to thank the Reviewer for taking the time to evaluate our work and for sharing their feedback. We agree with the Reviewer that it would be very interesting to reconstitute hundreds of kilobases or potentially megabases of chromatin. However, as we explain in more detail below, this is technically very challenging and currently not feasible. Our smaller-scale reconstitution has allowed us to demonstrate for the first time that it is possible to reconstitute chromatin domain formation and analyze features of domain boundaries *in vitro*. This has

allowed us to study the mechanisms that drive 3D genome folding from a new, different perspective. Our approach has therefore enabled us to make a number of observations about the relationship between nucleosome positioning and higher-order genome folding into chromatin domains in yeast, which we think are of general interest to the field.

COMMENTS

PLASMIDS: It is puzzling to me that the authors would choose to use plasmids containing 10kb of yeast DNA. First, the 3D structure of a ~10-20kb plasmid clearly has no relevance to a large long linear chromosome (I know they sometimes linearize the plasmid, but this is still not relevant and the maps look different from circular plasmids). Second, many 3D interactions tend to occur across scales including tens to hundreds of kb, even in yeast. But using tiny DNA segments, they make it impossible to observe the interactions that make 3D architecture interesting.

We agree with the Reviewer that we have not explained in sufficient detail why we have chosen to use a genomic plasmid library with ~10 kb inserts as DNA template for our chromatin reconstitutions. This reconstitution approach, based on generation of chromatin during salt gradient dialysis (SGD), is currently the only system with which it is possible to reconstitute *in-vivo*-like chromatin, with similar nucleosome densities as found *in vivo* and with regularly spaced and phased nucleosome arrays (Krietenstein et al, Cell 2016; Oberbeckmann et al, Nature Communications 2021). This system uses relatively small plasmids, since this facilitates efficient assembly of *in-vivo*-like chromatin. However, it is important to stress that ~10 kb chromatin regions are sufficiently large to study chromatin domain organization in yeast interphase, since these domains are on average ~5 kb in size (Hsieh et al, Cell 2015). As a result, we are able to capture multiple domains on individual ~10 kb fragments, as shown in Figure 2 in the manuscript. Our current set-up is therefore informative to study higher-order genome folding in yeast. To further clarify this point, we have generated larger-scale contact matrices (derived from the Micro-C data published in Hsieh et al, Nature Methods 2016), which show the *in vivo* organization of a few regions of 80-100 kb in yeast interphase. These matrices are shown in the figure below.

This figure shows that domains in yeast form at a scale of ~5 kb and that there are few interactions spanning >10 kb. Although the Reviewer is correct that we cannot examine interactions between multiple domains or long-range interactions that occur in yeast on mitotic chromosomes, our current set-up allows us to analyze the nature of individual chromatin domains and their boundaries in interphase, which is the focus of our study.

Since our work shows that nucleosome positioning plays an important role in the formation of chromatin domains, it is important that reconstituted chromatin closely resembles *in vivo* chromatin. Although we fully agree with the Reviewer that it would be very interesting to determine the 3D structure of larger reconstituted regions of chromatin, it is important to note that it is technically very challenging to reproducibly reconstitute larger pieces of chromatin with *in-vivo*-like features. Since this is not established at the moment, we have therefore chosen to work with SGD reconstitution of plasmids.

The reviewer is correct to point out that these plasmids are circular for the majority of the

analyses that we have performed. However, we show that the 3D conformation of circular chromatin is very similar compared to chromatin that has been linearized by BamHI digestion, except for the regions surrounding the BamHI cut sites, where we observe additional boundaries (as expected). We have included additional individual and "meta" contact matrices to highlight the similarity between chromatin organization on circular and linearized plasmids more clearly, which are shown in Extended Data Figure 4 and copied below.

We would like to thank the Reviewer for pointing out that we had insufficiently motivated the choice for our reconstitution system. We have therefore rewritten our Results section to clarify this (Lines 101-112). In addition, we have included a paragraph in the Discussion to explicitly discuss the limitations of our reconstitution system and our study (Lines 424-442). We have copied the key sections from these paragraphs below for convenience.

Lines 101-112:

*To this end, we adapted a previously developed system to reconstitute *S. cerevisiae* chromatin in vitro, which is based on the assembly of nucleosomes by salt gradient dialysis (SGD)^{21,22}. As a*

DNA template, we used a genomic plasmid library covering *S. cerevisiae* chromosomes V-IX. Each of these plasmids contains a ~7 kbp backbone and an insert covering a fraction of the *S. cerevisiae* genome with an average length of ~10 kbp. Incubation of this plasmid library with purified recombinant *S. cerevisiae* histone octamers (Extended Data Fig. 1a) in high-salt conditions and overnight dialysis into a low-salt buffer leads to spontaneous formation of nucleosomes. This reconstitution system has the important benefit that it allows for the generation of chromatin with a high nucleosome density that resembles *in vivo* chromatin²³. To further facilitate the formation of high nucleosome densities *in vitro*, we used negatively-supercoiled plasmid DNA amplified in *E. coli*, as negative supercoiling is thought to propagate nucleosome assembly during salt gradient dialysis^{24,25}.

Lines 424-442:

The SGD reconstitution system that we have used in this study has the advantage that it allows for reconstitution of chromatin with *in-vivo-like* features using a native, chromosome-wide DNA template²³. However, this system also has limitations. Most importantly, the reconstituted chromatin fibers are restricted in size, as they are generated from a genomic plasmid library with ~10 kbp inserts (Fig. 1a). This system is therefore limited to analyzing features of chromatin organization on a scale of ~10 kbp and cannot be used (in its current form) to analyze larger-scale features of genome folding. However, since the yeast genome is characterized by short-range interactions and folds into relatively small (~5 kbp) domains in interphase¹⁹, the SGD reconstitution system is suitable to study the formation of chromatin domains and the features of their boundaries in yeast. Furthermore, since Abf1 and Reb1 are the only transcription factors that are present in the reconstitution reactions, this system is limited to analyzing regions that are bound by these transcription factors and not informative for *in vitro* analysis of regions that are bound by other transcription factors *in vivo*. Another limitation of our study is that our analyses of the organization of reconstituted chromatin are solely based on 3C data. 3C is based on chromatin crosslinking, digestion, and proximity ligation, which is reflected in the resulting interaction patterns. These patterns may therefore be biased towards detecting interactions between regions that are strongly bound by proteins⁵¹. It would therefore be of interest to combine future reconstitution studies with microscopy-based approaches to measure the organization and dynamics of chromatin.

CHROMATIN DOMAINS: The authors repeatedly refer to “chromatin domains” without defining this term. In the 3D architecture field, there is already too many terms like TADs, insulated neighborhood, loop domain, contact domain, compartment, hub, tether, etc. It is not clear what a chromatin domain is. Please explicitly define it and say if/how this is relevant to other common terms like TAD, compartment, or loop domain.

We fully agree with the Reviewer that the nomenclature in the field is confusing and have therefore added a paragraph to the Introduction of our manuscript to clarify our choice for the term "chromatin domain" and its definition (Lines 40-60). In addition, we have added a section to the Discussion to clarify how chromatin domains in yeast relate to TADs and compartments (Lines 484-489). We have copied these sections below.

Lines 40-60:

At a larger scale, eukaryotic genomes organize into self-interacting domains. In mammals, these domains are formed by at least two distinct mechanisms¹⁰. First, active and inactive regions of chromatin form functionally distinct compartments that span a wide range of sizes¹¹. Second, a process of loop extrusion, mediated by Structural Maintenance of Chromosomes

(SMC) proteins and CTCF, organizes the genome into local structures termed Topologically Associating Domains (TADs), which usually range from 100 kbp to 1 Mbp in size^{12,13}.

*The higher-order organization of the genome into self-interacting domains is conserved in eukaryotes with smaller genomes, including *D. melanogaster*¹⁴ and *S. cerevisiae*^{15,16}, in which domain sizes range from 10-500 kbp and 2-10 kbp, respectively. These domains are usually referred to with the general terms chromatin domain, chromosomal domain, or chromosomal interaction domain (CID)^{14,16}. This reflects that the nature of the domains in these species and the mechanisms by which they are formed are less well understood. Hereafter, we will therefore adopt the general term chromatin domain to refer to these domains. Because the boundaries of chromatin domains in fly^{14,17,18} and yeast^{16,19} frequently overlap with promoters of highly transcribed genes, it has been proposed that the process of transcription or the transcriptional state of chromatin are key determinants of chromatin organization. There is currently no conclusive evidence for cohesin-mediated loop extrusion during G1 interphase in these species, but it is possible that this process also contributes to the basic organization of their genomes.*

Lines 484-489:

However, the notion that loop extrusion may not be required for the organization of the yeast genome during interphase is consistent with low levels of cohesin occupancy on chromatin during interphase⁶⁷ and with cohesin perturbation studies in yeast⁶³⁻⁶⁶. Together, this indicates that the chromatin domains that form in yeast interphase are not comparable to vertebrate TADs, which form via a loop extrusion process mediated by SMC proteins^{68,69}, and may be more similar to compartments.

HUMAN 3D ARCHITECTURE: The authors speculate on relevance to human 3D architecture and include the sentence "This demonstrates that neither loop extrusion nor transcription are

required for domain formation.”. Relevant to the above this is quite misleading and confusing in my opinion – the domains, TADs measuring hundreds of kb, that are thought to form through loop extrusion clearly have no relevant relationship with 10kb plasmids and it seems misleading to indicate so.

We apologize for the confusion, as we did not mean to imply that loop extrusion and transcription are not required for the formation of *any* type of chromatin domain across eukaryotes. However, we do believe that the observation that we can reconstitute *in-vivo*-like chromatin domains in absence of SMC proteins and the transcription machinery suggests that loop extrusion and transcription are not required for the formation of basic chromatin domains in yeast *per se*. The Reviewer is correct though that we cannot generalize these findings to larger domains in higher eukaryotes. In the previous version of the manuscript, we had already acknowledged that there is very convincing evidence that shows that loop extrusion plays an important role in genome organization in higher eukaryotes (vertebrates), by driving the formation of Topologically Associating Domains. Following the advice from the Reviewer, we have further clarified throughout the text (incl. abstract) that our work only directly informs about chromatin domains in yeast interphase and that any implications for genome organization in higher eukaryotes are of more speculative nature. This includes the paragraphs in the Discussion in which we discuss the role of transcription (Lines 468-478) and

loop extrusion (Lines 480-489) in the formation of chromatin domains. We have copied the key sections of these paragraphs below.

Lines 468-478:

*Although the relatively simple organization of the yeast genome is not directly representative of the more complex organization of mammalian genomes, it is interesting to point out that transcription and nucleosome positioning are similarly intertwined in mammals. We therefore speculate that it is possible that observations that have been interpreted as a driving role for transcription in genome folding in mammals could also reflect associated patterns of nucleosome positioning. Examples include the enrichment of active promoters at domain boundaries^{61,62}, which are also characterized by strong NFRs, as well as the interpretation of transcription perturbation experiments, which are generally also associated with changes in nucleosome positioning⁵². Although speculative, our findings therefore suggest that the role of transcription in genome organization *in vivo* may, to some extent, be mediated by the influence of transcription on nucleosome positioning.*

Lines 480-489:

*Since we were able to reconstitute *in-vivo*-like chromatin domains in absence of SMC proteins,*

our experiments also indicate that loop extrusion is not per se required for basic domain organization of the yeast genome in interphase. It should be noted though that cohesin-dependent loop extrusion is thought to play an important role in the regulation of yeast chromatin structure during S-phase and mitosis⁶³⁻⁶⁶. However, the notion that loop extrusion may not be required for the organization of the yeast genome during interphase is consistent with low levels of cohesin occupancy on chromatin during interphase⁶⁷ and with cohesin perturbation studies in yeast⁶³⁻⁶⁶. Together, this indicates that the chromatin domains that form in yeast interphase are not comparable to vertebrate TADs, which form via a loop extrusion process mediated by SMC proteins^{68,69}, and may be more similar to compartments.

Reviewer #3:

Remarks to the Author:

I have reviewed this manuscript previously for a different journal, and the same comments are shared here, as no relevant changes were made to address the previously shared concerns.

In the manuscript titled "In vitro reconstruction of chromatin domains", Quililan, Oberbeckmann et al. investigate the mechanisms important for the folding of chromatin in *S. cerevisiae*. The authors developed a strategy to reconstitute the chromatin in vitro and adapted the MNase-based 3C protocol to map its 3D conformation. By incubating plasmids that contain parts of the yeast genome with different purified chromatin remodelers (i.e. INO80, ISW2, Chd1) and general transcription factors (i.e. Abf1 and Reb1) they show that the regular spaced and phased nucleosomes are sufficient for the formation of chromatin domains, and that these are similar to those found in vivo. Furthermore, they show that the compaction of the domains depends on the size of the nucleosome linker which varied with the chromatin remodeller.

The line of research pioneered in this manuscript is highly promising for the development of its field, but its strong claims are currently not convincingly supported. The following major comments must be addressed before publication.

We would like to thank the Reviewer for their constructive and helpful feedback on our work. As explained in detail below, we have performed the suggested additional experiments and analyses to strengthen our conclusions.

Major comments:

- It has been previously shown that binding of multiple transcription factors can drive insulation.

The authors make the assumption that Abf1 and Reb1 bind *in vitro* based on the *in vivo* ChIP-seq data, but the manuscript does not directly demonstrate that Abf1 and Reb1 TFs are actually able to bind DNA in the *in vitro* conditions chosen, in the absence of chromatin remodellers. To address this, ChIP-seq experiments are necessary for both Abf1 and Reb1 in the different *in vitro* conditions tested (in presence of INO80, ISW2, Chd1 and general transcription factors). Furthermore, to strengthen the claim that nucleosome spacing is the main driver of chromatin domain formation, it is also necessary that assays without the transcription factors are performed and the results included in the manuscript as controls. The correlations between TF factor enrichment and the boundary strengths shown in Extended Data Fig 4a also need to be calculated based on the actual experimental *in vitro* ChIP-seq data that is currently missing.

We would like to thank the Reviewer for these helpful suggestions. We had not yet performed *in vitro* ChIP-seq experiments, since the appearance of nucleosome-free regions at binding sites for Abf1 and Reb1 suggests that these transcription factors bind to DNA in the *in vitro* conditions in presence of remodelers. However, we agree with the Reviewer that it is important to confirm the binding of the transcription factors *in vitro* in presence of remodelers and to test if the transcription factors also bind in absence of the remodelers. We therefore performed

ChIP-seq experiments for both Abf1 and Reb1 in all experimental *in vitro* conditions. The resulting data are shown in Extended Data Figure 2, which we have copied below. These data clearly show that Abf1 and Reb1 bind to DNA, both in presence and in absence of chromatin remodelers.

Following the suggestion from the Reviewer, we have re-analyzed the correlation between transcription factor enrichment and boundary strength in Extended Data Figure 4A using the *in vitro* ChIP-seq data (Extended Data Figure 6 in the revised manuscript; copied below). In the revised figure, we have included plots based on both the *in vivo* and the *in vitro* ChIP-seq data, so that the *in vivo* and *in vitro* Micro-C data can easily be compared across all conditions. We have also included additional analyses to further clarify the association between transcription factor binding and boundary formation.

In the previous version of the manuscript, we had included control conditions in which we incubated the reconstituted chromatin with the transcription factors Abf1 and Reb1 but without chromatin remodelers. Following the suggestion from the Reviewer, we have now also included *in vitro* Micro-C analyses of chromatin incubated with chromatin remodelers but without transcription factors as additional controls. In addition, we performed MNase-seq experiments in these conditions. These data are shown in Extended Data Figure 1 (MNase-seq analyses), Extended Data Figure 5 (individual contact matrices) and Figure 3 ("meta" contact matrices; copied below).

These experiments show clearly that inclusion of INO80, Chd1 and ISW2 in absence of transcription factors results in regular nucleosome spacing (evident in the "meta" contact matrices), but does not lead to the formation of phased arrays surrounding NFRs at transcription

factor binding sites (evident in the MNase-seq analyses). As expected, we therefore do not observe the formation of chromatin domains in the absence of the transcription factors.

These experiments highlight that both regular nucleosome spacing and nucleosome phasing surrounding NFRs (and not nucleosome spacing alone) are required for the formation of chromatin domains. Please note that this is consistent with our conclusions, as we state that "regularly spaced and phased nucleosome arrays surrounding NFRs" are required and sufficient for chromatin domain formation (see e.g. title of Figure 2). We apologize for any potential confusion regarding these findings/conclusions. We have reworded sections throughout the manuscript to further clarify this (see e.g. Lines 308-311).

- The main claim of the manuscript is that: "the formation of regular and phase nucleosome arrays is sufficient to reconstitute native chromatin domains" and it is concluded that transcription is not essential for the formation of chromatin domains in yeast. Nonetheless, many biochemical processes such as transcription, DNA repair etc., have an impact on the regular nucleosome positioning. Binding of the transcription machinery and transcription itself have been previously shown to impact chromatin condensation status and implicitly on nucleosome spacing. The authors focused their analysis on specific loci, enriched for Abf1 and Reb1 binding motifs, and did not address directly the function of transcription or other cellular biochemical processes on nucleosome spacing. Furthermore, there is evidence from work on mammalian chromatin, that transcription is important for boundary formation in some cases and not essential in others. To understand whether transcription in yeast might play a similar role the authors would need to look at other genomic areas and/or devise in vitro experiments that address this matter directly by for example studying chromatin configuration with and without the transcription machinery. Therefore, it is essential that broad claims about transcription not being important for domain formation are removed or correctly rephrased to be clearly made specific to the in vitro system used, without unsupported generalisation.

The Reviewer is correct that we have not directly assessed the function of transcription in chromatin domain formation in yeast. Having said that, we do believe that the observation that we can reconstitute *in-vivo*-like chromatin domains in absence of the transcription machinery suggests that transcription is not required for the formation of chromatin domains in yeast *per se*. However, the Reviewer is correct that we cannot generalize these findings directly to higher eukaryotes. In addition, we completely agree with the Reviewer that our findings do not mean that transcription does not influence chromatin domain formation or 3D genome folding more broadly. We fully agree that this is important to emphasize, especially since there is ample evidence that transcription can influence nucleosome positioning in yeast and thereby likely impact on

chromatin domain formation. We had already rewritten the corresponding sections of the Discussion to rephrase our interpretations and conclusions when we received the Reviewer's report following our previous journal submission. We have now further clarified the Discussion and removed any unsupported generalizations. This is most evident in Lines 444-478, which we have copied below for convenience.

Lines 444-478:

The observation that regular nucleosome positioning drives the formation of in-vivo-like chromatin domains in yeast has several implications that are relevant for the ongoing debate about the relationship between genome structure and function. For example, an important open question is whether transcription is responsible for the formation of chromatin domains or whether chromatin domains are formed first to enable transcription afterwards. Since domain boundaries overlap with active gene promoters and boundary strength scales with increased RNA polymerase II binding in yeast^{16,19}, it has been suggested that transcription may have a driving role in the organization of the yeast genome. However, it is important to note that transcription and nucleosome positioning are closely connected, since wide NFRs are a prerequisite for formation of the pre-initiation complex (PIC) and PIC occupancy scales with increased NFR width⁵²⁻⁵⁴. Based on in vivo observations, it is therefore difficult to disentangle the role of transcription and nucleosome positioning in 3D genome organization. The strength of our in vitro approach is that we can uncouple these two processes. Although we have not directly investigated the function of transcription, the observation that we can reconstitute in-vivo-like chromatin domains in absence of the transcription machinery suggests that transcription is not required for domain formation in yeast per se. This is in agreement with computer simulations that have shown that nucleosome positioning alone can predict domain organization of yeast interphase chromosomes⁵⁵. Notably, transcription has also been implicated in the regulation of chromatin compaction, as previous studies in yeast have reported that active transcription is associated with higher compaction¹⁶. However, it is possible that this connection between transcription and chromatin compaction is mediated by associated changes in nucleosome spacing, since highly transcribed genes generally have short linkers⁵⁶, which may be mediated by the recruitment of Chd1 to actively transcribed genes^{29,57-59}. Transcription is also thought to play a role in the 3D organization of the genome of higher eukaryotes⁶⁰. Although the relatively simple organization of the yeast genome is not directly representative of the more complex organization of mammalian genomes, it is interesting to point out that transcription and nucleosome positioning are similarly intertwined in mammals. We therefore speculate that it is possible that observations that have been interpreted as a driving role for transcription in genome folding in mammals could also reflect associated patterns of nucleosome positioning. Examples include the enrichment of active promoters at domain boundaries^{61,62}, which are also

characterized by strong NFRs, as well as the interpretation of transcription perturbation experiments, which are generally also associated with changes in nucleosome positioning⁵². Although speculative, our findings therefore suggest that the role of transcription in genome organization in vivo may, to some extent, be mediated by the influence of transcription on nucleosome positioning.

- The manuscript includes examples of regions that are enriched for the transcription factors investigated. To address the role of transcription factors in shaping chromatin structure, it is important to also investigate regions that are not enriched for these particular transcription factors. How does the conformation of domains not bound by the transcription factors investigated here compare to in vivo? If such investigations cannot be made, then the conclusions and interpretations must be down-toned to report on the specific system that was used, acknowledging that a lot more work is necessary before generalisations can be made.

Our experiments show that regularly spaced and phased nucleosome arrays that form around occupied Abf1 and Reb1 binding sites drive the formation of chromatin domains in *in vitro* chromatin, which look very similar to *in vivo* chromatin domains. We have focused our analyses and comparisons between *in vivo* and *in vitro* chromatin conformation on regions that contain Abf1 and Reb1 binding sites, since these transcription factors are the only transcription factors that are present in our *in vitro* reconstitutions. We do not expect that the conformation of *in vivo* chromatin regions that are enriched for binding of other transcription factors will look similar to the conformation of the *in vitro* chromatin, since the reconstituted chromatin has not been incubated with these transcription factors and therefore will not contain regularly spaced and phased nucleosome arrays in these regions. We therefore expect that regions without binding sites for Abf1 and Reb1 look similar in their conformation to the experimental conditions in which we add the remodelers without the transcription factors, (which does not lead to domain formation, as shown in Extended Data Figure 5). This is indeed the case, as shown in the Figure that we created for the Reviewer below.

We agree with the Reviewer that it is important to clarify the limitations of the reconstitution system that we have used and avoid over-generalization. We have therefore added a paragraph to the Discussion to clarify the limitations of our study (Lines 424-442), which we have copied below. We have also toned down our conclusions and extensively rewritten the Discussion accordingly.

Lines 424-442:

The SGD reconstitution system that we have used in this study has the advantage that it allows for reconstitution of chromatin with in-vivo-like features using a native, chromosome-wide DNA template²³. However, this system also has limitations. Most importantly, the reconstituted chromatin fibers are restricted in size, as they are generated from a genomic plasmid library with ~10 kbp inserts (Fig. 1a). This system is therefore limited to analyzing features of chromatin organization on a scale of ~10 kbp and cannot be used (in its current form) to analyze larger-scale features of genome folding. However, since the yeast genome is characterized by short-range interactions and folds into relatively small (~5 kbp) domains in interphase¹⁹, the SGD reconstitution system is suitable to study the formation of chromatin domains and the features of their boundaries in yeast. Furthermore, since Abf1 and Reb1 are the only transcription factors that are present in the reconstitution reactions, this system is limited to analyzing regions that are bound by these transcription factors and not informative for in vitro analysis of regions that are bound by other transcription factors in vivo. Another limitation of our study is that our analyses of the organization of reconstituted chromatin are solely based on 3C data. 3C is based on chromatin crosslinking, digestion, and proximity ligation, which is reflected in the resulting interaction patterns. These patterns may therefore be biased towards detecting interactions

between regions that are strongly bound by proteins⁵¹. It would therefore be of interest to combine future reconstitution studies with microscopy-based approaches to measure the organization and dynamics of chromatin.

Other comments

- It is essential to explicitly clarify whether circular or linearised Micro-C data were used for most of the analysis. The matrices shown on Fig1,d suggest that regions surrounding BamHI restriction sites still form chromatin contacts after the digestion. It is important to acknowledge this observation in the manuscript and comment on whether there is incomplete DNA digestion in the sample or another problem.

We would like to thank the Reviewer for bringing this up. To assess and improve the digestion efficiency, we analyzed the fragment size of an individual plasmid from the plasmid library after remodeling and digestion with different BamHI concentrations. We find that the concentration that we had used previously did indeed result in rather incomplete digestion. With a higher BamHI concentration, we can achieve more complete digestion. This is shown in Extended Data Figure 4a, which we have pasted below. Although not every individual BamHI restriction site is digested, we estimate that >90 % of the plasmids are linearized after digestion with a higher BamHI concentration. (Note that many plasmids contain multiple BamHI restriction sites, of which only one needs to be digested for the plasmid to be linearized.)

We have repeated the *in vitro* Micro-C experiments on reconstituted chromatin that has been linearized with a higher BamHI concentration. These data are shown in Figure 1d. We have also rewritten the corresponding paragraph to acknowledge that digestion is incomplete. We have pasted Figure 1d and the key section from this paragraph (Lines 204-213) below.

Lines 204-213:

As expected, we find that plasmid linearization creates additional interaction "boundaries" at the BamHI restriction sites, since the regions upstream and downstream of digested restriction sites are no longer adjacent but on opposite ends of the linearized plasmids (Fig. 1d). Note that there are still low-frequency interactions spanning these boundaries, as digestion is not complete (Extended Data Fig. 4a) and regions upstream and downstream of digested restriction sites are still connected on plasmids with one BamHI restriction site. However, importantly, linearization of the plasmids does not affect nucleosome positioning (Extended Data Fig. 4b) or patterns of higher-order nucleosome folding (Fig. 1d, Extended Data Fig. 4c- e). For all remaining experiments, we therefore used circular plasmids.

Please note that we had already mentioned in the previous version of our manuscript (in Lines 193-194) that we used circular plasmids for all remaining experiments. In the current version of our manuscript, this sentence is located in Lines 212-213 (copied above).

- To help understand how much can the work be generalised, the manuscript should provide further information about the region selected for Fig 2, namely its gene density, GC content, etc relative to the *S. cerevisiae* genome, to help understand to what extent the region is somehow representative.

We had already shown the gene density above the matrices in Figure 2. We have clarified this in the figure legend. Following the Reviewer's suggestion, we have also included information about the DNA sequence composition in the regions shown in Figure 2. Since the yeast genome is very AT-rich, especially in promoter regions (as opposed to mammalian genomes), we prefer to show the AT content in Figure 2, which indirectly also informs about GC content.

- Several studies have highlighted differences in chromatin organisation between ligation- and microscopy-based technologies, and suggest a more cautious interpretation of the results (for

example Williamson et al., 2014; DOI: 10.1101/gad.251694.114). The claims of the current manuscript are strong and should acknowledge that the results might be inherently biased by the cross-linking events. For instance, ligation-based technologies have a strong bias in capturing interactions that are strongly bound by proteins. It is therefore possible that the strength of the domains observed here is biased by the number of proteins cross-linked, and it cannot be excluded that the domains form and/or are weaker in the absence of TF binding.

The Reviewer is correct that our analyses are based on proximity ligation and that these techniques have inherent limitations. Following the Reviewer's suggestions, we have included a section in the Discussion to explicitly discuss these limitations and their implications. This section is described in Lines 424-442 (which is copied in this rebuttal letter in response to the last major comment of the Reviewer).

- It is recommended that the authors' included a statement about whether the proposed mechanisms are limited to 10kb chromatin domains or could potentially contribute to the formation of larger-scale chromatin domains such as mammalian TADs.

We agree with the Reviewer that this is important to discuss. We have included a section in the Discussion to clarify this in Lines 480-489, which we have copied below.

Lines 480-489:

Since we were able to reconstitute in-vivo-like chromatin domains in absence of SMC proteins, our experiments also indicate that loop extrusion is not per se required for basic domain organization of the yeast genome in interphase. It should be noted though that cohesin-dependent loop extrusion is thought to play an important role in the regulation of yeast chromatin structure during S-phase and mitosis⁶³⁻⁶⁶. However, the notion that loop extrusion may not be required for the organization of the yeast genome during interphase is consistent with low levels of cohesin occupancy on chromatin during interphase⁶⁷ and with cohesin perturbation studies in yeast⁶³⁻⁶⁶. Together, this indicates that the chromatin domains that form in yeast interphase are not comparable to vertebrate TADs, which form via a loop extrusion process mediated by SMC proteins^{68,69}, and may be more similar to compartments.

Decision Letter, first revision:

18th Oct 2023

Dear Marieke,

hope this email finds you well.

Your Article, "In vitro reconstitution of chromatin domains reveals a role for nucleosome positioning in 3D genome organization" has now been seen by 3 referees. You will see from their comments below that Reviewers #1 and #3 are satisfied with the revised version and do not have further requests. While Reviewer #2 has some remaining small points that we would like you to address. We are interested in the possibility of publishing your study in Nature Genetics and would like to consider your response to these concerns in the form of a revised manuscript before we make a final decision on publication. Please do not hesitate to get in touch if you would like to discuss these issues further.

We therefore invite you to revise your manuscript taking into account all Reviewer #2 points. Please highlight all changes in the manuscript text file. At this stage we will need you to upload a copy of the manuscript in MS Word .docx or similar editable format.

*2) If you have not done so already please begin to revise your manuscript so that it conforms to our Article format instructions, available

[here](http://www.nature.com/ng/authors/article_types/index.html).

*3) Include a revised version of any required Reporting Summary:

[redacted]

We hope to receive your revised manuscript within four to eight weeks. If you cannot send it within this time, please let us know.

Thank you!

My best wishes,
Chiara

Chiara Anania, PhD
Associate Editor
Nature Genetics
<https://orcid.org/0000-0003-1549-4157>

Referee expertise:

Referee #1:

Referee #2:

Referee #3:

Reviewers' Comments:

Reviewer #1:

Remarks to the Author:

I remain supportive of this study. It is timely and innovative and provides novel insight into mechanisms of chromatin domain formation.

Reviewer #2:

Remarks to the Author:

The authors have submitted a revised manuscript, which addresses many of the points raised by the three reviewers. On the presentation side of things, the authors do a much better job of not overclaiming in the revised manuscript, they for the most part define their terms, and they have toned down the comparisons to human 3D architecture, though I think they could have gone further on some of these points.

I do think the manuscript is interesting and as I mentioned in the first review, I like the bottom-up reconstitution approach. However, my concerns about 3D architecture largely remain for these small plasmid-based systems – I would view this paper as more relevant to nucleosome positioning than to 3D genome organization, along the lines of the bottom-up nucleosome positioning papers from Kevin Struhl and Oliver Rando et al. during the 2010s.

I have a few mostly quite minor points/questions remaining, but overall although I find the study of interest to nucleosome positioning, I remain only moderately enthusiastic about the 3D genome organization implications.

SMALL POINTS (these do not question the main conclusions, but would improve clarity)

1. please include yeast or *S Cerevisiae* in the manuscript title, to make the type of chromatin domains studied in the manuscript more clear.
2. The authors chose the examples in Fig 2 based on Abf1 and Reb1 enrichment, can they show a similar figure with in vivo and in vitro panels for regions without Abf1 and Reb1 enrichment, perhaps as a supplementary figure.
3. The discussion around line 323-324 is confusing, since RSC has the longest NFR but weakest insulation, therefore saying that there is a quantitative relationship does not seem totally fair. Where do the data points in Fig 3f come from and can you label with which remodeler. Why does Fig3f seem inconsistent with 3e and 4a.
4. Can the authors be more specific about the quantitative relationship in 4b? is it a power-law (then please tell us the exponent) or is it exponential?
5. Lines 491-498 discuss how CTCF organizes "local" chromatin and mediated insulation without cohesin. Can the authors please define "local" (e.g. 100 bp, 1kb or 10kb?) to improve clarity?

Reviewer #3:

Remarks to the Author:

We thank the authors for their extensive efforts to answer our concerns. The current version of the manuscript is very well written and the additional work fortifies the claims of the manuscript. We do not have any additional comments and therefore recommend the manuscript to be published in Nature Genetics.

Author Rebuttal, first revision:

Reviewer #1:

Remarks to the Author:

I remain supportive of this study. It is timely and innovative and provides novel insight into mechanisms of chromatin domain formation.

We thank the Reviewer for their support and helpful feedback.

Reviewer #2:

Remarks to the Author:

The authors have submitted a revised manuscript, which addresses many of the points raised by the three reviewers. On the presentation side of things, the authors do a much better job of not overclaiming in the revised manuscript, they for the most part define their terms, and they have toned down the comparisons to human 3D architecture, though I think they could have gone further on some of these points.

I do think the manuscript is interesting and as I mentioned in the first review, I like the bottom-up reconstitution approach. However, my concerns about 3D architecture largely remain for these small plasmid-based systems – I would view this paper as more relevant to nucleosome positioning than to 3D genome organization, along the lines of the bottom-up nucleosome positioning papers from Kevin Struhl and Oliver Rando et al. during the 2010s.

I have a few mostly quite minor points/questions remaining, but overall although I find the study of interest to nucleosome positioning, I remain only moderately enthusiastic about the 3D genome organization implications.

We thank the Reviewer for their helpful feedback and have addressed their remaining concerns in detail below.

SMALL POINTS (these do not question the main conclusions, but would improve clarity)

1. please include yeast or *S. cerevisiae* in the manuscript title, to make the type of chromatin domains studied in the manuscript more clear.

We thank the Reviewer for this suggestion. While we understand the reasoning of the Reviewer, we think that adding “yeast/*S. cerevisiae*” to the title could be distracting and is not necessary, since we already mention “yeast/*S. cerevisiae*” twice in the abstract. We have discussed this

with the Editor and agreed to keep the title as it is for now and follow the guidance from the Editorial Team on this matter during the pre-acceptance checks.

2. The authors chose the examples in Fig 2 based on Abf1 and Reb1 enrichment, can they show a similar figure with *in vivo* and *in vitro* panels for regions without Abf1 and Reb1 enrichment, perhaps as a supplementary figure.

We thank the Reviewer for this suggestion. We have included a comparison of *in vivo* and *in vitro* Micro-C data for regions without Abf1 and Reb1 enrichment in Extended Data Figure 5 (pasted below), to which we refer in Lines 251-254. Please note that we do not expect that the *in vitro* conformation of chromatin regions that are not bound by Abf1 and Reb1 look similar to the conformation of *in vivo* chromatin, since the reconstituted chromatin has not been incubated with the other transcription factors that bind to *in vivo* chromatin in these regions. We therefore expect that regions without binding sites for Abf1 and Reb1 look similar in their conformation to the experimental conditions in which we add the remodelers without the transcription factors (which does not lead to domain formation, as shown in Extended Data Figure 6). This is indeed the case. Please also note that we mention in the paragraph in the Discussion in which we discuss the limitations of our study that our current reconstitution system is only suitable to analyze regions bound by Abf1 and Reb1 (Lines 441-444).

3. The discussion around line 323-324 is confusing, since RSC has the longest NFR but weakest insulation, therefore saying that there is a quantitative relationship does not seem totally fair. Where do the data points in Fig 3f come from and can you label with which remodeler. Why does Fig3f seem inconsistent with 3e and 4a.

We agree with the Reviewer that this section/figure is somewhat confusing.

Although RSC mediates the formation of a wide NFR, it does not have regular spacing activity. As a result, *in vitro* chromatin incubated with RSC does not form chromatin domains. We therefore do not include RSC in our assessment of the relationship between NFR width and insulation strength, since RSC does not mediate the formation of insulated domains. We have clarified this in the corresponding paragraphs (Lines 321-341), which we have pasted below.

Lines 321-341:

These data confirm that regular nucleosome positioning is required for the formation of chromatin domain boundaries, since chromatin incubated with remodelers that create regular nucleosome arrays (INO80, Chd1, and ISW2) forms strong domain boundaries, whereas incubation with RSC does not result in the formation of regular nucleosome arrays and clear domain boundaries. The MNase-seq data also show that of the three remodelers with spacing activity, INO80 forms the widest NFR, followed by Chd1 and ISW2. This indicates that boundary strength is dependent on the size of the NFR established at transcription factor binding sites. To confirm this, we calculated the distance between the nucleosome borders and transcription factor binding sites (Fig. 3e), in order to determine the width of the NFR (Fig. 3f). We find that INO80, Chd1 and ISW2 create NFRs of distinct sizes, which vary between 126, 92, and 68 bp in width, respectively. Plotting the average boundary strength against these NFR sizes indicates that insulation strength scales proportionally with NFR width in chromatin incubated with remodelers with regular spacing activity (Fig. 3f).

Consistent with the literature^{22,26,28}, we find that RSC does not drive the formation of regular nucleosome arrays, but does generate very wide NFRs (Fig. 3d,e). We find that this results in a chromatin interaction pattern characterized by a depletion of signal at the NFR itself and weak insulation at the NFR. We therefore conclude that insulation strength is dependent on both the width of the NFR at the chromatin domain boundary and the formation of a regularly spaced nucleosome array, phased to the boundary site.

Figure 3f shows the NFR widths according to the schematic drawing in the left panel of Figure 3e. This panel shows two smaller arrows that indicate the distance between the nucleosome border and the transcription factor (which is directly derived from the data) and a larger arrow that indicates the width of the NFR (which is calculated based on the distance between the

nucleosome border and the transcription factor). The “Distance to TF / bp” is plotted in Figure 3e and the “Length of NFR / bp” is plotted in Figure 3f. We have clarified this in the text and figure legends. We have also clarified in the legend of Figure 3f which data points correspond to which remodeler.

Figure 3e and 3f focus on the sizes of the NFRs, whereas Figure 4a focuses on linker lengths. This is indicated in the schematic drawing in the top panel of Figure 4a by the arrows and indicated in the graph (“Linker length / bp”). We have clarified this in the legend of Figure 4a. Since the NFR widths and linker lengths vary across the chromatin remodelers and are not directly correlated, the values in Figure 3e/f and Figure 4a are different.

4. Can the authors be more specific about the quantitative relationship in 4b? is it a power-law (then please tell us the exponent) or is it exponential?

We thank the Reviewer for noticing that this was not specified. We plotted \log^{10} interaction frequencies in these distance decay curves, as commonly used in the field (see e.g., Krietenstein et al. *Molecular Cell* 2020, Figure S1E; Ohno et al. *Cell* 2019, Figure S1F). We have clarified this in the legend of Figure 4b (and corresponding Extended Data Figure 9a).

5. Lines 491-498 discuss how CTCF organizes “local” chromatin and mediated insulation without cohesin. Can the authors please define “local” (e.g. 100 bp, 1kb or 10kb?) to improve clarity?

We have clarified this in Lines 499-501 (pasted below).

Lines 499-501:

We have previously reported that CTCF mediates local insulation (up to 10-20 kb) in mammalian genomes in a cohesin-independent manner³⁸.

Reviewer #3:

Remarks to the Author:

We thank the authors for their extensive efforts to answer our concerns. The current version of the manuscript is very well written and the additional work fortifies the claims of the manuscript. We do not have any additional comments and therefore recommend the manuscript to be published in *Nature Genetics*.

We thank the Reviewer for their support and helpful feedback.

Decision Letter, second revision:

2nd Nov 2023

Hi Marieke,

hope this email finds you well.

Thank you for submitting your revised manuscript "In vitro reconstitution of chromatin domains reveals a role for nucleosome positioning in 3D genome organization" (NG-A62290R2). We carefully reviewed your point-to-point response to Reviewers and revised manuscript. We are satisfied with how you addressed all Reviewers' comments; therefore, we'll be happy in principle to publish it in Nature Genetics, pending minor revisions to comply with our editorial and formatting guidelines.

Thank you again for your interest in Nature Genetics. Please do not hesitate to contact me if you have any questions.

Congratulations!

Sincerely,
Chiara

Chiara Anania, PhD
Associate Editor
Nature Genetics
<https://orcid.org/0000-0003-1549-4157>

Final Decision Letter:

15th Dec 2023

Dear Dr. Oudelaar,

I am delighted to say that your manuscript "In vitro reconstitution of chromatin domains shows a role for nucleosome positioning in 3D genome organization" has been accepted for publication in an

upcoming issue of Nature Genetics.

Your paper will be published online after we receive your corrections and will appear in print in the next available issue. You can find out your date of online publication by contacting the Nature Press Office (press@nature.com) after sending your e-proof corrections.

Please note that *Nature Genetics* is a Transformative Journal (TJ). Authors may publish their research with us through the traditional subscription access route or make their paper immediately open access through payment of an article-processing charge (APC). Authors will not be required to make a final decision about access to their article until it has been accepted. [Find out more about Transformative Journals](https://www.springernature.com/gp/open-research/transformative-journals)

Authors may need to take specific actions to achieve [compliance](https://www.springernature.com/gp/open-research/funding/policy-compliance-faqs) with funder and institutional open access mandates. If your research is supported by a funder that requires immediate open access (e.g. according to [Plan S principles](https://www.springernature.com/gp/open-research/plan-s-compliance)) then you should select the gold OA route, and we will direct you to the compliant route where possible. For authors selecting the subscription publication route, the journal's standard licensing terms will need to be accepted, including <https://www.nature.com/nature-portfolio/editorial-policies/self-archiving-and-license-to-publish>. Those licensing terms will supersede any other terms that the author or any third party may assert apply to any version of the manuscript.

If you have not already done so, we invite you to upload the step-by-step protocols used in this manuscript to the Protocols Exchange, part of our on-line web resource, natureprotocols.com. If you complete the upload by the time you receive your manuscript proofs, we can insert links in your article that lead directly to the protocol details. Your protocol will be made freely available upon publication of your paper. By participating in natureprotocols.com, you are enabling researchers to more readily reproduce or adapt the methodology you use. [Natureprotocols.com](http://natureprotocols.com) is fully searchable, providing your protocols and paper with increased utility and visibility. Please submit your protocol to <https://protocolexchange.researchsquare.com/>. After entering your [nature.com](http://www.nature.com) username and password you will need to enter your manuscript number (NG-A62290R3). Further information can be found at <https://www.nature.com/nature-portfolio/editorial-policies/reporting-standards#protocols>

Sincerely,
Chiara

Chiara Anania, PhD
Associate Editor
Nature Genetics
<https://orcid.org/0000-0003-1549-4157>